# A hierarchical transcriptional network activates specific CDK inhibitors that regulate G2 to control cell size and number in Arabidopsis

Yuji Nomoto[1,11], Hirotomo Takatsuka[1,11], Kesuke Yamada[1], Toshiya Suzuki[2], Takamasa Suzuki [3], Ying Huang[4], David Latrasse[4], Jing An [4], Magdolna Gombos[5], Christian Breuer[6], Takashi Ishida [6,10], Kenichiro Maeo[7], Miyu Imamura[7], Takafumi Yamashino[7], Keiko Sugimoto [6,8], Zoltán Magyar[5], László Bögre[9], Cécile Raynaud[4], Moussa Benhamed[4] & Masaki Ito [1✉]

How cell size and number are determined during organ development remains a fundamental question in cell biology. Here, we identified a GRAS family transcription factor, called SCARECROW-LIKE28 (SCL28), with a critical role in determining cell size in Arabidopsis. SCL28 is part of a transcriptional regulatory network downstream of the central MYB3Rs that regulate G2 to M phase cell cycle transition. We show that SCL28 forms a dimer with the AP2-type transcription factor, AtSMOS1, which defines the specificity for promoter binding and directly activates transcription of a specific set of SIAMESE-RELATED (SMR) family genes, encoding plant-specific inhibitors of cyclin-dependent kinases and thus inhibiting cell cycle progression at G2 and promoting the onset of endoreplication. Through this dose-dependent regulation of *SMR* transcription, SCL28 quantitatively sets the balance between cell size and number without dramatically changing final organ size. We propose that this hierarchical transcriptional network constitutes a cell cycle regulatory mechanism that allows to adjust cell size and number to attain robust organ growth.

---

[1] School of Biological Science and Technology, College of Science and Engineering, Kanazawa University, Kakuma-machi, Kanazawa 920-1192, Japan. [2] National Institute of Genetics, 1111 Yata, Mishima, Shizuoka 411-8540, Japan. [3] College of Bioscience and Biotechnology, Chubu University, Kasugai, Aichi 487-8501, Japan. [4] Université Paris-Saclay, CNRS, INRAE, Univ Evry, Institute of Plant Sciences Paris-Saclay (IPS2), 91405 Orsay, France. [5] Institute of Plant Biology, Biological Research Centre, Szeged 6726, Hungary. [6] RIKEN Center for Sustainable Resource Science, Yokohama 230-0045, Japan. [7] Graduate School of Bioagricultural Sciences, Nagoya University, Furocho, Chikusa-ku, Nagoya 464-8601, Japan. [8] Department of Biological Sciences, Graduate School of Science, The University of Tokyo, 7-3-1 Hongo, Bunkyo-ku, Tokyo 113-0033, Japan. [9] Centre for Systems and Synthetic Biology, Department of Biological Sciences, Royal Holloway University of London, Egham TW20 0EX, UK. [10]Present address: Faculty of Advanced Science and Technology, Kumamoto University, Kumamoto 860-8555, Japan. [11]These authors contributed equally: Yuji Nomoto, Hirotomo Takatsuka ✉email: masakito@se.kanazawa-u.ac.jp

During organ growth and development, cell proliferation is intricately controlled in space and time. Governed by developmental programs[1–3] and influenced by active plant responses to environmental conditions[4,5], the number of cells produced during organ growth is set by the cell cycle speed during proliferation and the point when cells exit to cellular differentiation. Controlled cell proliferation requires coordinately regulated gene expression within and upon exit from the cell cycle. Generally, there are two main groups of genes showing waves of transcription during the cell cycle; G1/S-specific and G2/M-specific genes[6]. The G1/S-specific genes facilitate initiation and progression of DNA replication and are typically regulated by the activity of E2F transcription factors[7,8]. Generally, E2F dimerizes with Dimerization Partner (DP) proteins to activate or repress their target genes, depending on their association with the Retinoblastoma-Related (RBR) repressor protein[9]. On the other hand, most G2/M-specific genes are positively or negatively regulated by MYB3R family of transcription factors in plants[10,11]. Some members of MYB3Rs are specifically expressed during G2/M and act as transcriptional activators, while others act as transcriptional repressors of G2/M-specific genes[12–14]. Recent studies identified the two main groups of transcription factors, E2Fs and MYB3Rs, which had been studied separately, as part of the same multi-protein complex in Arabidopsis[11]. This E2F-MYB3R complex is evolutionarily related to the DREAM (DP, Retinoblastoma-like, E2F, and MuvB) complex reported in *Drosophila* and human cells[11,15,16]. The metazoan DREAM complex plays a predominant role in repressing both G1/S- and G2/M-specific genes, thus promoting cell cycle exit and maintaining cellular quiescence[17,18]. The DREAM complex in Arabidopsis shows significant differences from metazoan complexes, which include involvement of plant-specific subunits and existence of diversified complexes with different subunit compositions[9,11,15].

Transcriptional regulation during cell cycle generally constitutes multi-layered hierarchical networks, in which master regulators regulate other transcription factors, which further regulate each other or downstream genes[19–21]. Notably, studies in yeasts showed that cell cycle transcriptional activators that function during one stage of the cell cycle regulate transcriptional activators that function during the next stage, forming a connected regulatory network that is itself a cycle[22]. However, in plants, such a hierarchical network composed of cell cycle transcription factors, E2F and MYB3R, has not yet been uncovered. Exploring the transcription network during cell cycle may uncover missing important factors and hidden mechanisms governing plant-specific cell cycle regulation.

In this work, we identified a mitosis specific GRAS family transcription factor, called SCARECROW-LIKE28 (SCL28). In accordance with our work, a recent report identified the same genes being directly regulated by MYB3Rs[23]. Here, we demonstrate that SCL28 acts in association with the AP2-type transcription factor, AtSMOS1, to directly activate transcription of a specific set of *SMR* family genes, encoding plant-specific cyclin-dependent kinase (CDK) inhibitors[24]. This regulatory network inhibits the G2 to M phase transition during the cell cycle and promotes the onset of endoreplication, an atypical cell cycle consisting of repeated DNA replication without mitosis[4]. Our study identified a G2/M regulatory pathway that controls cell cycle length, and is likely to be important in optimizing cellular functions by setting cell size and number during organ growth.

## Results

### Identification of a mitotic GRAS-type transcription factor.
By analyzing transcripts specific in mitotic cells, we have identified an Arabidopsis GRAS family transcription factor that we

designated *E1M*[25]. A recent report looking for mitotic genes in the root meristem uncovered the same gene called *SCL28*[23], the name hereafter also adopted in this study. In a synchronized culture of Arabidopsis MM2d cells, we showed that this gene exhibited G2/M-specific transcript accumulation, which closely resembles that of the mitotic cyclin *CYCB1;2* (Fig. 1a). As for most G2/M-specific genes, the so-called mitosis-specific activator (MSA) element that serves as a binding motif for MYB3Rs[10,26] were repeatedly present within the proximal promoter regions of *SCL28* (Fig. 1b). Binding of MYB3R to the *SCL28* promoter is supported by our published data from chromatin immunoprecipitation (ChIP) with MYB3R3 followed by high-throughput sequencing (ChIP-Seq; Fig. 1b)[11] and DNA affinity purification sequencing (DAP-Seq) data reported for MYB3R1, MYB3R4, and MYB3R5 (Supplementary Fig. 1a)[27]. In addition, loss of MYB3R activators (*myb3r1/4* double mutant) or MYB3R repressors (*myb3r1/3/5* triple mutant) resulted in significant down- or up-regulation of *SCL28*, respectively (Fig. 1c). GUS reporter activity driven by *SCL28* promoter decreased significantly in the *myb3r1/4* double mutant and became essentially undetectable when the MSA elements in the *SCL28* promoter had been deleted (Fig. 1d). Collectively, these observations support the idea that *SCL28* is a direct target of both activator and repressor type MYB3Rs in Arabidopsis. Similar to the G2/M-specific CYCB1;1-GFP accumulation[28], we observed patchy pattern of SCL28-GFP signal from a construct driven by native *SCL28* promoter in root meristem, suggesting cell cycle-regulated protein accumulation (Fig. 1e). Taken together, as has been recently shown[23], *SCL28* is a mitotic gene directly regulated by MYB3Rs.

To evaluate whether MYB3Rs act as part of DREAM complex on *SCL28* transcription, we searched lists of target genes bound by potential DREAM components, RBR and TESMIN/TSO1-LIKE CXC 5 (TCX5), as defined by ChIP-Seq experiments reported previously[15,29]. This examination revealed that TCX5, but not RBR, showed a significant association to *SCL28* locus in vivo. However, *SCL28* showed no significant change in expression in neither *rbr, tcx5 tcx6* double, nor *e2fa e2fb e2fc* triple mutants, indicating that *SCL28* transcription depends exclusively on MYB3Rs (Supplementary Fig. 1b).

### SCL28 strongly affects cell size.
To analyze the biological function of SCL28, we generated transgenic plants overexpressing *SCL28* under the strong *RPS5A* promoter (proRPS5A::SCL28). These plants, herein designated *SCL28*[OE], showed general growth retardation both in seedling and adult stages (Fig. 2a). Cell size in these plants was significantly enlarged in all examined organs and tissues, such as root tip, leaf mesophyll, embryo, and inflorescence stem (Fig. 2b and Supplementary Fig. 2). An increased cell size was apparent in both post-mitotic differentiated cells and proliferating cell populations in root and shoot apical meristems (Fig. 2c and Supplementary Fig. 2). As described later, the increased cell size in *SCL28*[OE] leaves was associated with elevated levels of cellular ploidy induced by enhanced endoreduplication. The loss-of-function *scl28* mutant (Supplementary Fig. 3) showed an opposite cellular phenotype to *SCL28*[OE], having cells with significantly reduced size (Fig. 2b, c and Supplementary Fig. 2). However, unlike *SCL28*[OE], the overall stature of *scl28* mutant plants was largely normal and indistinguishable from that of wild type (WT) plants (Fig. 2a).

To explore the developmental origin of cell size differences, we performed kinematic analysis on growing first leaf pairs by monitoring size and number of palisade cells (Fig. 2d–g). This analysis showed that cells in *scl28* leaves were smaller than WT already during initial stages of organ development when most cells are actively proliferating (Fig. 2d). During this early

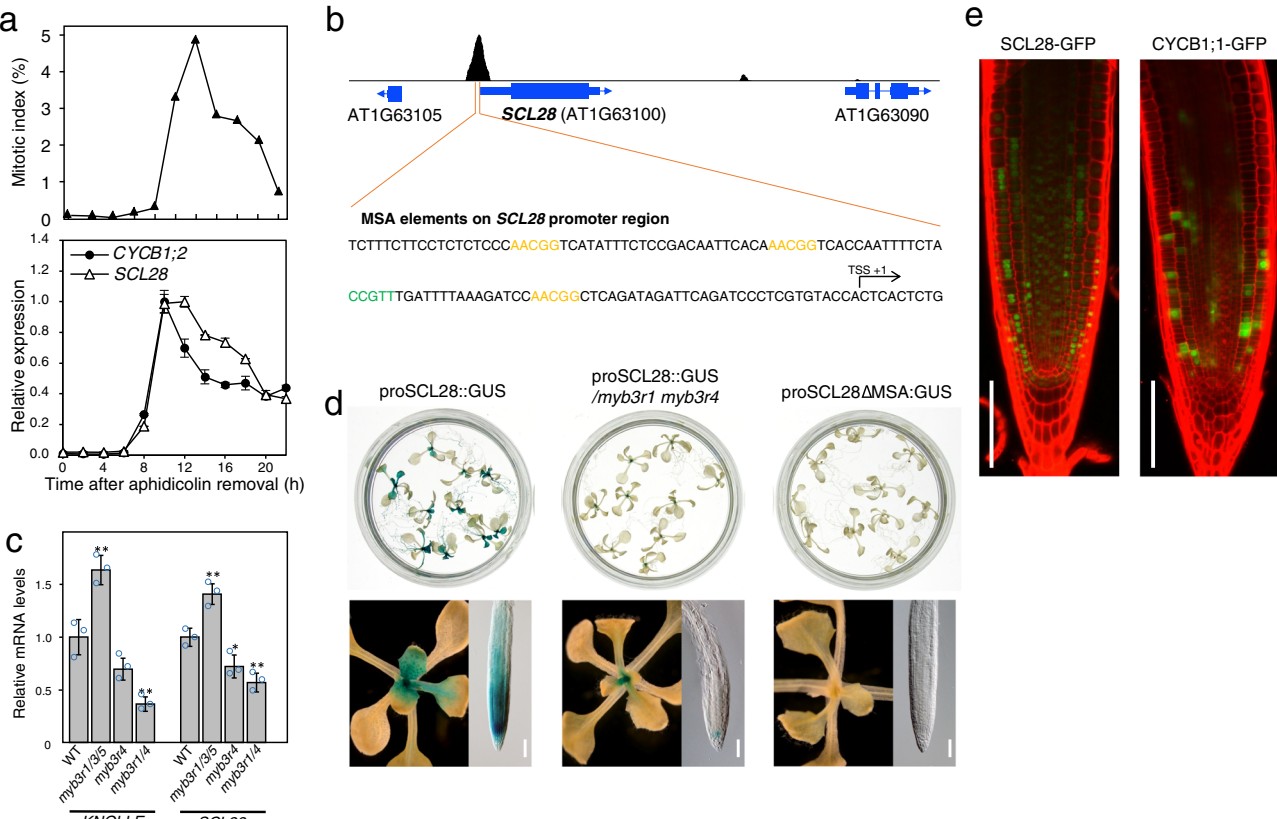

**Fig. 1 SCL28 expression is cell cycle-regulated under the control of MYB3R transcription factors. a** G2/M-specific accumulation of *SCL28* transcript. Arabidopsis MM2d cells were synchronized by aphidicolin treatment and used for qRT-PCR analysis to quantify *SCL28* mRNA levels (lower). Synchronous progression of cell cycle was monitored by measuring mitotic index (upper). As a representative of G2/M-specific genes, *CYCB1;2* was also analyzed in the same way for comparison. Expression data are shown as averages from three technical replicates (±SD). **b** MYB3R3 binds to the upstream region of *SCL28*. The ChIP-Seq profile of MYB3R3 indicates its direct binding to the proximal promoter region of *SCL28*. The nucleotide sequence around the ChIP-Seq peak contains MSA elements repeated four times, of which core motifs (AACGG) are shown by orange (forward orientation) and green (reverse orientation) letters. **c** Levels of *SCL28* transcripts are significantly affected by mutations in *myb3r* genes. Transcript levels of *SCL28* and *KNOLLE* were analyzed by qRT-PCR in plants 12 days after sowing (DAS) that carry mutations of *myb3r* genes in the indicated combinations. Expression data were normalized to *ACT2* and shown as averages from three biological replicates (±SD). Statistical significance was determined using two-sided Student's *t* test. *$P < 0.05$, **$P < 0.01$. **d** Promoter activity of *SCL28* requires the presence of MYB3R activators and MSA elements. GUS reporter expression was analyzed in wild type (WT) and *myb3r1/4* plants carrying proSCL28::GUS and WT plants carrying proSCL28ΔMSA::GUS, in which all MSA elements in the *SCL28* promoter were mutated. GUS staining around shoot apex and root tip regions is shown at higher magnification in the lower panels. Multiple independent lines for each construct showed similar difference in GUS staining. Scale bar indicates 100 μm. **e** Patchy pattern expression of SCL28-GFP protein. Root meristems of proSCL28::SCL28-GFP and proCYCB1;1::CYCB1;1-GFP plants were analyzed by confocal laser scanning microscopy (CLSM) after counterstaining of the cell wall with propidium iodide (PI). Green and red signals indicate fluorescence of GFP and PI, respectively. Similar expression patterns of SCL28-GFP were confirmed in multiple independent lines. Scale bar indicates 100 μm.

proliferative stage, increasing cell number was also more rapid in *scl28* than WT, consistent with a higher cell division rate in *scl28* leaves compared with WT (Fig. 2e, f). Conversely, duration of cell proliferation remained largely unchanged between *scl28* and WT (Fig. 2e, f). Therefore, the increased cell number in *scl28* leaves is due to accelerated cell division, rather than increased duration of cell proliferation. When leaf area was compared, however, *scl28* and WT showed no clear difference throughout the course of leaf development (Fig. 2g). In *scl28* leaves, accelerated cell division was balanced by reduced cell size, thus maintaining total organ size unchanged during leaf development. The cellular effect of *SCL28*^OE was generally opposite to the *scl28* mutant. The number of palisade cells per leaf was significantly reduced in *SCL28*^OE plants (Fig. 2e) due to severely inhibited cell proliferation (Fig. 2f). Although dramatic cell enlargement partially counteracted the reduced cell number, leaf area was still reduced in *SCL28*^OE compared with WT (Fig. 2d, g). We also analyzed cell size and number of epidermal pavement cells during leaf development in

*scl28* and *SCL28*^OE lines and found the response of this cell type to altered SCL28 activity was comparable to what we have shown for palisade cells (Supplementary Fig. 4).

In the root tip, cell size difference was similarly apparent among *SCL28*^OE, *scl28*, and WT (Fig. 2c, h, i). This cell size difference was maintained along the apical-basal axis of the root from cells adjacent to the quiescent center to those at the transition zone, as well as in differentiated cells at distal positions (Fig. 2h). Therefore, the behavior of cells in the root meristems of *SCL28*^OE and *scl28* was generally consistent with those observed in developing leaves.

**Effects of SCL28 on mitotic cell cycle and endocycle.** To directly explore the role of SCL28 in cell cycle progression, we performed live-cell imaging of root meristem cells after introgression of the PCNA-GFP cell cycle marker into *SCL28*^OE and *scl28* mutant. The PCNA-GFP line allows quantification of cell cycle stages

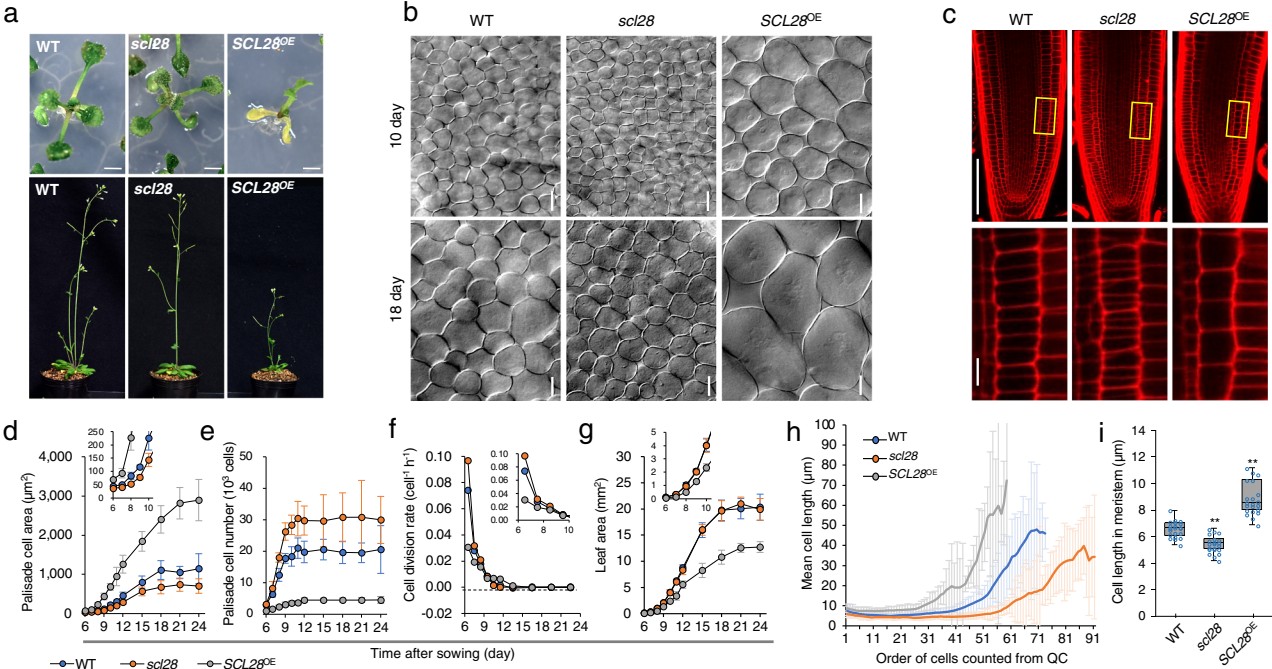

**Fig. 2 SCL28 strongly affects cell size. a** Macroscopic phenotypes caused by loss- and gain-of-function of *SCL28*. WT, *scl28*, and *SCL28*[OE] plants were photographed at 11 (upper) and 30 (lower) DAS. Scale bar in upper panels indicates 10 mm. **b** Cell size analysis of leaf palisade tissue from WT, *scl28*, and *SCL28*[OE] plants. First leaf pairs from plants at 10 or 18 DAS were cleared and analyzed by differential interference contrast (DIC) microscopy. Scale bar indicates 20 µm. **c** Cell size analysis of root meristem from WT, *scl28*, and *SCL28*[OE] plants at 5 DAS. Primary roots were stained with PI, and analyzed by CLSM. Regions surrounded by yellow rectangles are shown at a higher magnification in the lower panels. Scale bars indicate 100 µm (upper) and 10 µm (lower). **d–g** Kinematic analysis of leaf growth in WT, *scl28*, and *SCL28*[OE] plants. Time-course changes of palisade cell area (**d**), number of palisade cells per leaf (**e**), cell division rate (**f**), and leaf area (**g**) were analyzed in developing first leaf pairs. Values are averages from the data of leaves from ten different plants (±SD), in each of which more than 20 cells were analyzed. Inset shows the same data with an expanded y-axis scale. **h** Spatial distribution of cell size across root meristems in WT, *scl28*, and *SCL28*[OE] plants at 5 DAS. Mean cell length was calculated at each position along the cortical cell file, which is defined by counts of cortical cells from the quiescent center. Values are averages from the data analyzed in 21 roots from different plants (±SD), in each of which more than 30 cells were analyzed. **i** Quantitative analysis of cortical cell length within root meristems. Boxplot was generated using data collected from roots of 21 different plants, in each of which more than 20 cells were analyzed. In boxplot, 'boxes' represent the interquartile range (IQR), midline indicates the median, and whiskers extend to the largest and smallest observations within 1.5 × IQR. Statistical significance compared with WT was determined using two-sided Student's *t* test. **P < 0.01.

based on its intracellular fluorescent patterns[30]. During G1, PCNA-GFP appears as uniform fluorescence within the whole nuclei, which alters into dotty and speckled nuclear signals during early and late S phase, respectively. During G2, fluorescence again becomes uniform within the nuclei, then disappears upon onset of mitosis and remains undetectable until the exit from mitosis. By our definition, the G1 period begins with re-appearance of nuclear fluorescence after mitosis and ends with emergence of dotty GFP signals, whereas the G2 period begins with conversion of dotty into uniform nuclear PCNA-GFP signals and ends with disappearance of this nuclear fluorescence. Based on these definitions, we analyzed data from live-cell imaging and calculated the average duration of each cell cycle phase in WT root meristem cells to be 5.1 h in G1, 2.5 h in S, and 11.3 h in G2 (Fig. 3a, b). G2/M duration was oppositely affected in the *scl28* and *SCL28*[OE] lines, shortened 19% in the former and lengthened 33% in the later. In contrast, G1 length showed only limited alteration, but became longer in the *scl28* mutant (Fig. 3a, b), which is consistent with the existence of an additional cell size checkpoint at G1/S that compensates for shortened G2[31]. This suggests that SCL28 actively inhibits progression through G2 and prevents entry into mitosis. In agreement with the role of SCL28 in G2 duration of the mitotic cell cycle, our ploidy analysis of developing leaves showed that SCL28 also affects endoreplication (Fig. 3c, d, and Supplementary Fig. 5). During leaf development, *SCL28*[OE] plants initiated earlier endoreplication, indicated by 8 C cell emergence

as early as 8 days after sowing (DAS) and thereafter consistently showing higher cellular ploidy levels compared with WT. Elevated ploidy levels were associated with dramatically increased cell size in *SCL28*[OE] leaves as shown earlier (see Fig. 2b, d). In the *scl28* mutant, ploidy levels were not affected during early stages of leaf development and only showed a modest decrease at 20 DAS. During the earliest stage (8 DAS) before onset of endoreplication, we observed a smaller proportion of 4 C cells in *scl28* compared with WT (Fig. 3c), which supports the cell cycle analysis, showing reduced G2 duration (Fig. 3a). In summary, our data suggest that SCL28 has a role to inhibit the G2 to M phase transition.

**SCL28 acts together with AtSMOS1 as a heterodimer.** Phylogenetic analysis of GRAS family proteins indicated that SCL28 constitutes a unique clade together with SMOS2 in rice (Supplementary Fig. 6a). Rice *smos2* was initially identified as a mutant showing reduced organ size with component cells smaller than those in WT[32]. Another rice mutant, *smos1*, with a phenotype similar to that in *smos2*, has loss-of-function mutation in a unique gene encoding an AP2-type transcription factor[33]. Because a physical interaction between SMOS1 and SOMS2 has been reported[32], we postulated that SCL28 may act through interaction with an Arabidopsis protein orthologous to SMOS1. When AP2-type transcription factors were phylogenetically analyzed, we found At2g41710 to be the likely Arabidopsis ortholog

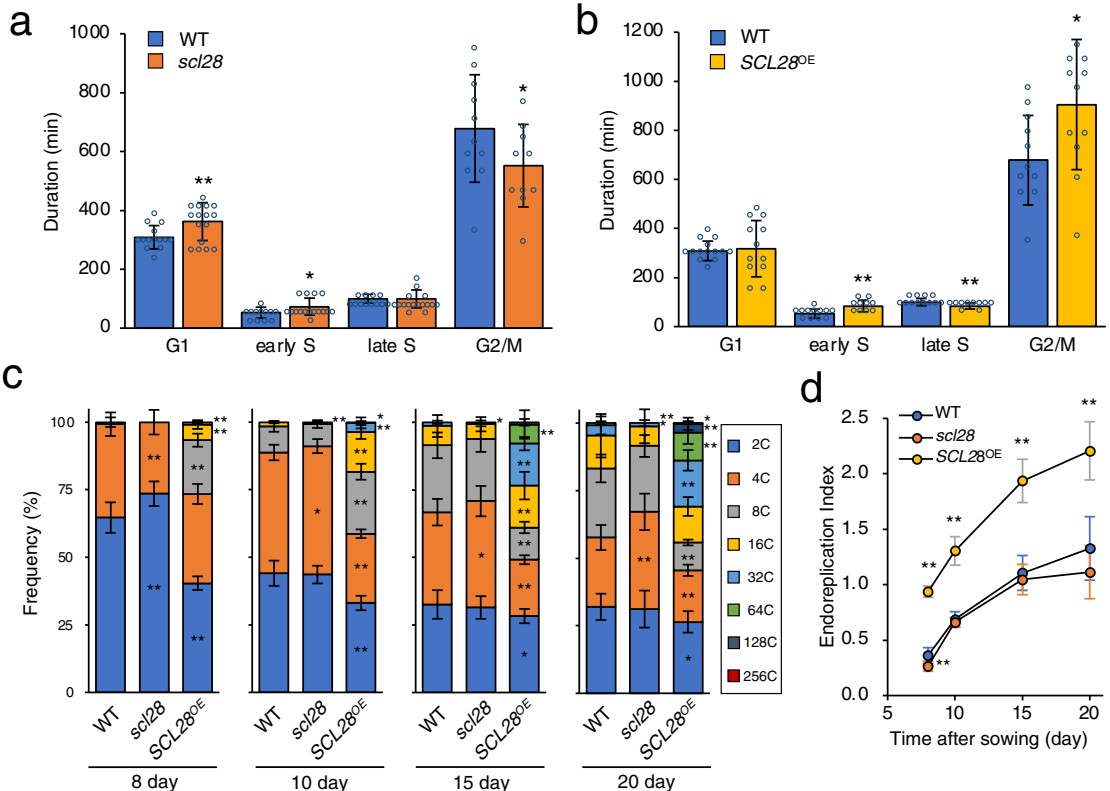

**Fig. 3 SCL28 inhibits G2 progression and induces endoreplication. a** G2 duration is shortened in *scl28*. Epidermal cells in root meristems of WT, *scl28*, and *SCL28*[OE] plants carrying the PCNA-GFP marker were analyzed by live-cell imaging for measuring length of G1, early S, late S, and G2/M phases in the cell cycle. Data are shown as averages (±SD). Numbers of biological replicates for calculating length of G1, early S, late S, and G2/M, respectively, are 14, 12, 12 and 11 for WT, and 16, 16, 16 and 10 for *scl28*. Statistical significance compared with WT was determined using one-sided Student's *t* test. *$P < 0.05$, **$P < 0.01$. **b** Prolonged G2 duration in *SCL28*[OE] plants. Cell cycle analysis in the root meristem was performed as in (**a**). Data are shown as averages (±SD). Numbers of biological replicates for calculating length of G1, early S, late S, and G2/M, respectively, are 14, 12, 12 and 11 for WT, and 12, 10, 10, and 12 for *SCL28*[OE]. Statistical significance was determined and shown as in (**a**). **c** Ploidy analysis of *scl28* and *SCL28*[OE] plants. First leaf pairs of WT, *scl28*, and *SCL28*[OE] plants were subjected to flow cytometric analysis to determine ploidy distribution during leaf development. Data are shown as averages from ten biological replicates (±SD). Statistical significance compared with WT was determined using two-sided Student's *t* test. *$P < 0.05$, **$P < 0.01$. **d** Time-course change in cellular ploidy levels during leaf development. Data presented in (**c**) were used for calculating endoreplication index, which represents mean number of endoreplication cycles per leaf cells. Data are shown as averages from ten biological replicates (±SD). Statistical significance compared with WT was determined and shown as in (**c**).

of SMOS1 (Supplementary Fig. 6b). To test the physical interaction between SCL28 and At2g41710, we performed yeast two-hybrid and bimolecular fluorescence complementation (BiFC) assays to obtain clear results showing that they indeed interact in yeast and in plant cells, respectively (Fig. 4a, b). We concluded that the observed protein-protein interaction is evolutionarily related to the SMOS1-SMOS2 interaction in rice and named At2g41710 as AtSMOS1.

To examine AtSMOS1 function, we analyzed a T-DNA insertion mutant for this gene (Supplementary Fig. 7) and found a small cell size phenotype that closely resembled that of *scl28* (Fig. 4c–f). However, the overall stature of *atsmos1* was largely indistinguishable from WT plants, as is the case for *scl28* (Supplementary Fig. 8a). To analyze epistasis between *scl28* and *atsmos1*, we performed genetic analysis by quantitatively comparing phenotypes of *scl28*, *atsmos1* and *scl28 atsmos1* double mutants. These single and double mutants showed essentially equivalent phenotypes in terms of palisade cell size (Fig. 4c, d) and cell size in the root meristem (Fig. 4e, f). This indicates that, as with SCL28, AtSMOS1 also impacts cell size. The lack of additive effect between *atsmos1* and *scl28* suggests that these proteins may act in the same pathway, which is consistent with the interaction between these proteins. We then

tested whether the enlarged cells in *SCL28*[OE] relies on the presence of AtSMOS1, and indeed in the *atsmos1* mutant background, the strong effects of *SCL28*[OE] on cell size of leaf palisades (Fig. 4g, h) and root meristem cells (Fig. 4i, j), as well as whole plant growth, completely diminished (Supplementary Fig. 8b). Collectively, these observations strongly suggest that SCL28 and AtSMOS1 function cooperatively to regulate cell size in different plant organs.

To analyze AtSMOS1 expression, we generated plants expressing AtSMOS1-GFP driven by its own promoter. These transgenic plants showed nuclear localization of AtSMOS1-GFP in various tissues and organs such as root meristem, developing leaves, and cotyledons (Fig. 4k). However, unlike SCL28, AtSMOS1-GFP was uniformly expressed in meristematic cells, suggesting a cell cycle-independent AtSMOS1 expression. Nonetheless, cells expressing AtSMOS1 and SCL28 were overlapping, supporting the idea that they interact to form a heterodimer in vivo. We also noted an E2F binding element in the *AtSMOS1* promoter, which binds RBR in published ChIP-Seq data[29]. We found that E2FB binds to *AtSMOS1* promoter in our ChIP-qPCR experiment and that *AtSMOS1* is significantly upregulated in the *e2fa e2fb e2fc* triple mutant (Supplementary Fig. 9), suggesting that *AtSMOS1* is indeed regulated by E2Fs. One explanation for

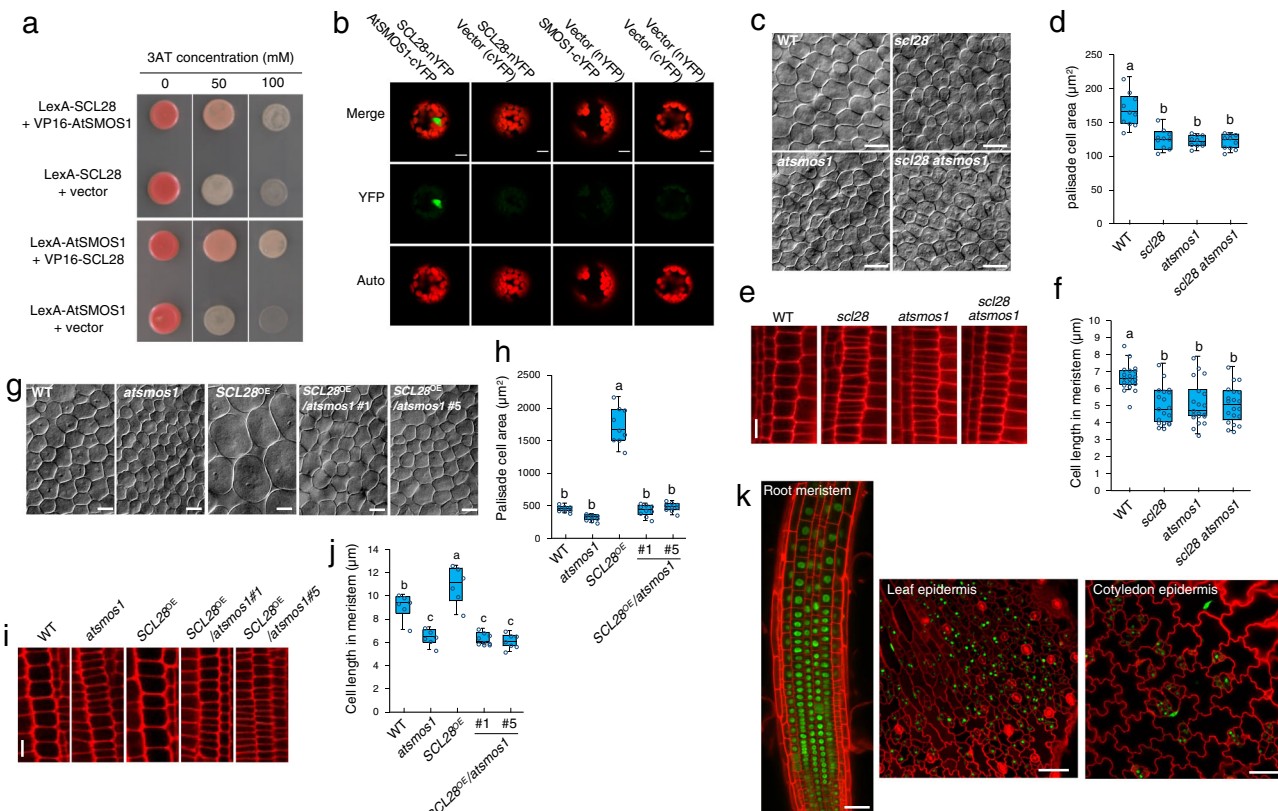

**Fig. 4 Formation of active heterodimer between SCL28 and AtSMOS1. a** SCL28-AtSMOS1 interaction in yeast two-hybrid assay. Yeast strains carrying two indicated constructs were grown on synthetic medium lacking His and containing indicated concentration of 3-amino-1,2,4-triazole (3-AT). **b** SCL28-AtSMOS1 interaction in bimolecular fluorescence complementation (BiFC) analysis. YFP fluorescence (YFP), autofluorescence of chloroplasts (Auto) and their merged image (Merge) were taken from mesophyll protoplasts that were transfected with the indicated plasmid constructs. Scale bars indicate 10 μm. The BiFC experiment was repeated twice with similar results. **c** Leaf palisade cells were observed by DIC microscopy using first leaf pairs from WT plants, and *scl28*, *atsmos1*, and double *scl28 atsmos1* mutants at 10 DAS. Scale bar indicates 20 μm. **d** Quantification of palisade cell area in first leaf pairs from plants with indicated genotypes. Boxplot was generated using data collected from leaves of ten different plants, in each of which more than 50 cells were analyzed (midline = median, box = IQR, whiskers = 1.5 × IQR). Different letters above boxplots indicate significant differences (*P* < 0.05) based on one-way ANOVA and Tukey's test. **e** Cortical cell files were observed by CLSM using PI-stained meristems of primary roots from WT plants, and *scl28*, *atsmos1*, and double *scl28 atsmos1* mutants at 5 DAS. Scale bars indicate 10 μm. **f** Quantification of cortical cell length in root meristems from plants with indicated genotypes. Boxplot was generated using data collected from roots of 20 different plants, in each of which more than 20 cells were analyzed (midline = median, box = IQR, whiskers = 1.5 × IQR). Different letters above boxplots indicate significant differences as in (**d**). **g** DIC observation of palisade cells from first leaf pairs of WT, *atsmos1*, *SCL28*^OE plants, and those possessing *atsmos1* and *SCL28*^OE in combination. Plants at 15 DAS were used. Scale bar indicates 20 μm. **h** Quantification of palisade cell area in first leaf pairs from the plants with indicated genotypes. Boxplot was generated using data collected from leaves of nine different plants, in each of which more than 30 cells were analyzed (midline = median, box = IQR, whiskers = 1.5 × IQR). Different letters above the boxplots indicate significant differences as in (**d**). **i** CLSM observation of PI-stained root meristems from WT, *atsmos1*, *SCL28*^OE plants, and those possessing *atsmos1* and *SCL28*^OE in combination. Plants at eight DAS were used for observation of cortical cell files. Scale bar indicates 10 μm. **j** Quantification of cortical cell length in root meristems of plants with indicated genotypes. Boxplot was generated using data collected from roots of multiple individual plants (midline = median, box = IQR, whiskers = 1.5 × IQR). Number of biological replicates for each line was 6, except for *SCL28*^OE/*atsmos1*#1 for which 9 biological replicates were analyzed. In each plant, more than 20 cells were measured. Different letters above boxplots indicate significant differences as in (**d**). **k** Accumulation patterns of AtSMOS1-GFP protein. Root meristem, and epidermis of leaf and cotyledon from plants at six DAS carrying proAtSMOS1::AtSMOS1-GFP were analyzed by CLSM after counterstaining of the cell wall with PI. Similar expression patterns of AtSMOS1-GFP were confirmed in multiple independent lines. Scale bars indicate 50 μm.

insignificant change of *AtSMOS1* mRNA in *rbr* and *tcx5 tcx6* might be that RBR- and DREAM-dependent regulation on E2Fs is developmental specific, and was not apparent at the seedling stage.

**Downstream targets of SCL28.** Studies of *smos1* and *smos2* rice mutants suggest a role in post-mitotic cell expansion through the regulation of the *Oryza sativa PHOSPAHATE INDUCED1* (*OsPHI-1*) gene[32]. Because we observed SCL28 expression specific to meristematic cells, we postulated that SCL28 should largely influence actively-proliferating cells before the onset of cell

expansion. To identify the downstream targets of SCL28, we analyzed genome-wide gene expression changes in *scl28*, *SCL28*^OE, and *atsmos1* by conducting RNA-sequencing (RNA-Seq) and microarray experiments (Supplementary Data 1). Considering the epistatic phenotypes among *scl28*, *SCL28*^OE, and *atsmos1* mutants, the critical downstream genes of SCL28 should be affected in *scl28* and *atsmos1* in a similar manner, as well as in *SCL28*^OE in an opposing manner. To reveal the genes satisfying these expression criteria, we conducted an overlapping analysis of differentially expressed genes in *scl28*, *atsmos1*, and *SCL28*^OE, and identified 21 genes that are significantly (adjusted *P* value < 0.05) downregulated in both *scl28* and *atsoms1*, and upregulated in

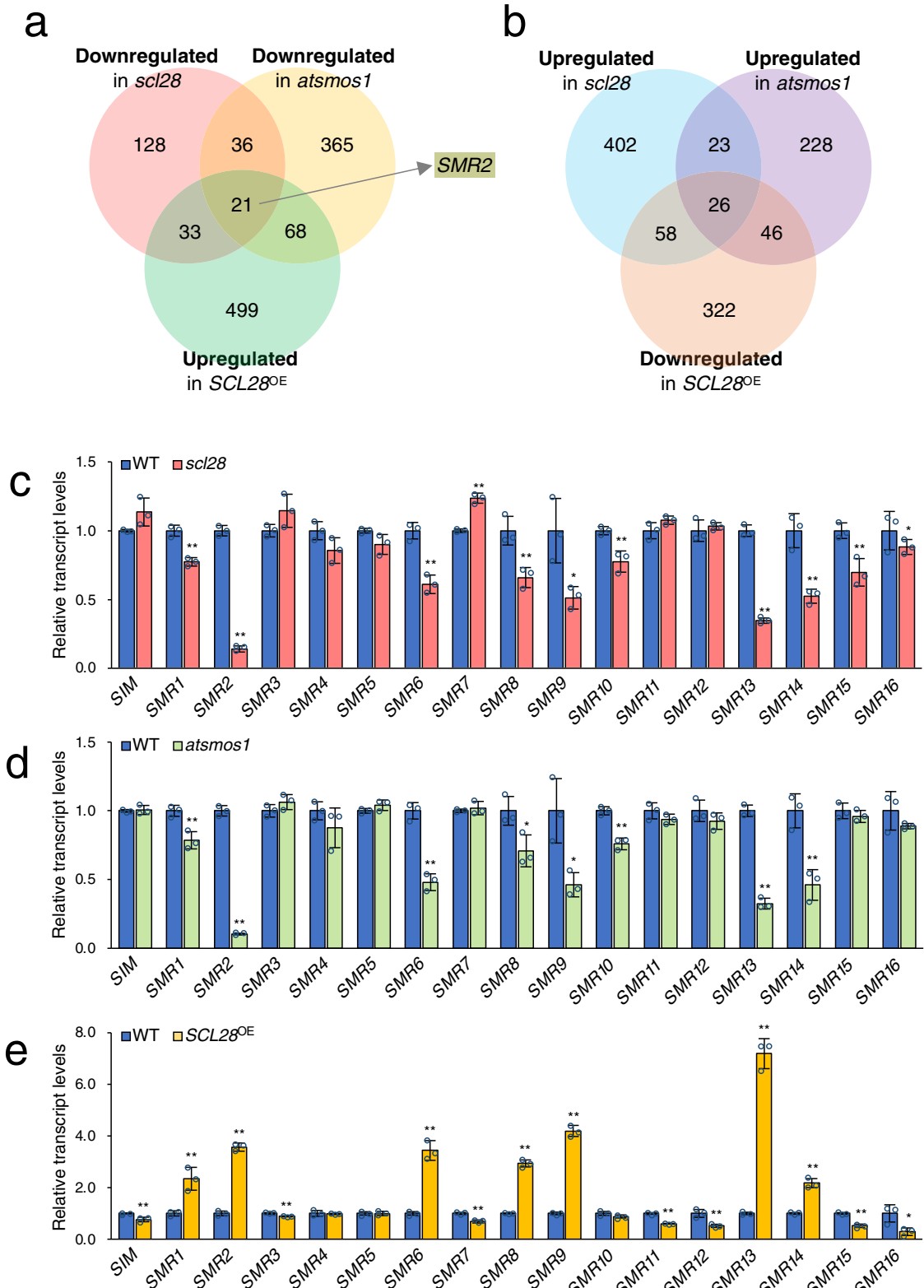

**Fig. 5 SCL28 and AtSMOS1 activate transcription of *SMR* genes. a** Venn diagram representing overlap between genes downregulated in *scl28* and *atsmos1*, and upregulated in *SCL28*^OE plants. The *SMR2* gene was found in the overlap of these three categories. Upregulated and downregulated genes were defined by a criterion of false-discovery rate (FDR)-adjusted *P* value < 0.05, which was obtained from expression data of three biological replicates. **b** Venn diagram representing overlap between genes upregulated in *scl28* and *atsmos1*, and downregulated in *SCL28*^OE plants. Upregulated and downregulated genes were statistically defined as in (**a**). **c**–**e** Transcript levels of *SMR* genes in *scl28* (**c**), *atsmos1* (**d**), and *SCL28*^OE (**e**) plants at 8 DAS. qRT-PCR analysis of all 17 genes in the SMR family was performed to compare expression levels with WT plants. Data are averages from three biological replicates (±SD). Statistical significance compared with WT was determined using two-sided Student's *t* test. *P < 0.05, **P < 0.01.

$SCL28^{OE}$ (Fig. 5a and Supplementary Fig. 10a), as well as 26 genes upregulated in both *scl28* and *atsoms1*, and downregulated in $SCL28^{OE}$ (Fig. 5b and Supplementary Fig. 10b). Among these genes, we focused on *SIAMESE-RELATED2* (*SMR2*), encoding a member of SMR family proteins, as proteins of this family represent plant-specific CDK inhibitors and some members, such as SIM and SMR1, positively affect cell size and negatively influence cell division, reflecting the observed effect of SCL28[1,24]. According to RNA-Seq and microarray data, in addition to *SMR2*, we found other *SMR* family members as candidates of downstream genes, with decreased expression levels in both *scl28* and *atsmos1* and increased levels in $SCL28^{OE}$. To further analyze *SMR* family genes, we conducted quantitative reverse-transcription PCR (qRT-PCR) analysis to examine expression of all members of this family in *scl28*, *atsmos1*, and $SCL28^{OE}$, and identified at least seven members (*SMR1*, *SMR2*, *SMR6*, *SMR8*, *SMR9*, *SMR13*, and *SMR14*) out of all 17 as potential target genes of SCL28 and AtSMOS1 (Fig. 5c–e). Consistent with this conclusion, we also observed reduced expression of proSMR2::SMR2-GFP and proSMR13::SMR13-GFP in root meristems in *scl28* or *atsoms1* mutant backgrounds compared with WT (Supplementary Fig. 11).

KIP-RELATED PROTEINs (KRPs) constitute another family of CDK inhibitors in addition to SMRs in plants. The general view is that KRP proteins primarily connect with CDKA to negatively regulate G1/S[34], while SMR proteins inhibit G2/M by associating with both CDKA and CDKB1[24,35]. There are seven *KRP* members encoded in the Arabidopsis genome, and some we did find deregulated in the *scl28*, *atsmos1*, and $SCL28^{OE}$ plants. However, in contrast to the *SMR*s, the change in expression of *KRP*s were generally moderate and inconsistent with the activating function of SCL28-AtSMOS1 heterodimer, and thus likely to be an indirect pleiotropic effect (Supplementary Fig. 12). Therefore, the SCL28 and AtSMOS1 may directly affect expression of *SMR* members, but not those of the *KRP* family, consistent with our results that SCL28 preferentially affects G2 duration in meristematic cells.

**Genome-wide mapping of SCL28 and AtSMOS1 binding sites**. In order to define direct targets for SCL28 and AtSMOS1, combined with transcriptome analysis, we performed ChIP-Seq assays using proRPS5A::SCL28-GFP and proAtSMOS1::AtSMOS1-GFP lines to identify their binding genomic loci. Consistent with our emphasis on downregulation of *SMR* genes, which suggests that SCL28 and AtSMOS1 act as transcriptional activators, the majority of binding sites are located in promoter regions proximal to transcription start sites (TSS) (Supplementary Fig. 13) and located within nucleosome-free and highly accessible chromatin regions (Supplementary Fig. 14a–d). Their enrichment levels at TSS correlated well with mRNA levels of corresponding genes, suggesting a positive correlation between their binding and transcriptional activity (Supplementary Fig. 14e, f).

The ChIP-Seq data analysis uncovered 463 and 4,287 genes as targets of SCL28 and AtSMOS1, respectively (Supplementary Data 2). Comparing these target genes revealed the presence of a significant overlap, a set of 308 common targets that accounts for 66% of SCL28 targets (Fig. 6a). On one hand, this result confirms our genetic analysis, suggesting that SCL28 function largely relies on the interaction with AtSMOS1. On the other hand, AtSMOS1 may show a wider range of functions both dependent on and independent of SCL28. This interpretation is also supported by our RNA-Seq data showing larger number of downregulated genes in *atsmos1* compared to *scl28* and these downregulated genes have a relatively small overlap (see Fig. 5a). We also found only a limited overlap between common targets identified by

ChIP-Seq and those that show regulated expression in transcriptome analysis of *scl28*, *atsmos1* and $SCL28^{OE}$ lines (Fig. 6b). Gene ontology (GO) enrichment analysis of the 308 common targets revealed overrepresented GO terms related to cell cycle, such as "regulation of mitotic nuclear division" and "regulation of DNA endoreduplication" (Supplementary Fig. 15). This overrepresentation of cell cycle-related GO terms largely relies on the presence of five common genes that belong to such GO categories. All these genes were found to be members of *SMR* family genes, *SMR2*, *SMR4*, *SMR6*, *SMR8*, and *SMR9*, all of which, except *SMR4*, showed significant downregulation in both *scl28* and *atsmos1* and upregulation in $SCL28^{OE}$ in our qRT-PCR experiments described above (see Fig. 5). Though three *SMR* genes—*SMR1*, *SMR13*, and *SMR14*—with significant expression changes in all lines failed to fulfill the criteria of ChIP-Seq data analysis, visual inspection revealed recognizable peaks for both SCL28 and AtSMOS1 in ChIP-Seq profiles, suggesting these *SMR*s are also direct common targets (Fig. 6c and Supplementary Fig. 16). The ChIP-Seq peaks were observed at either 5′ (*SMR1*, *SMR2*, and *SMR4*), 3′ (*SMR6*, *SMR9*, and *SMR14*) or both 5′ and 3′ (*SMR8* and *SMR13*) regions of the target *SMR* loci. In most cases, we found that the positions of ChIP-Seq peaks adjacent to *SMR* loci coincided for SCL28 and AtSMOS1, suggesting that they associate with the same sites. Our ChIP-Seq data also showed that none of the *KRP* genes were found as common targets of SCL28 and AtSMOS1, further confirming that SCL28-AtSMOS1 directly regulates some *SMR*s but not *KRP*s.

To find common binding sites and sequence motifs for SCL28 and AtSMOS1, we compared the precise positions of ChIP-Seq peaks along the gene structure. Dot-plot analysis of the common targets showed frequent clustering of SCL28 and AtSMOS1 peaks at the same positions relative to TSS and transcription end site (TES; Fig. 6d). Similar analysis at a genome-wide scale confirmed that SCL28 frequently targets the same sites as AtSMOS1 around the TSSs (Supplementary Fig. 17a). Within the common peaks for SCL28 and AtSMOS1, a DNA motif of C(a/t)T(a/t)GGATNC(c/t)(a/t) could be identified as an overrepresented *cis* element (Fig. 6e). This motif is indeed present among more than 40% of the common targets, but is much less frequent among targets of either SCL28 or AtSMOS1, which contain quite different *cis* elements based on sequence overrepresentation analysis (Supplementary Fig. 17b, c). This suggests that the formation of SCL28-AtSMOS1 heterodimer generates a specific sequence preference for DNA binding to regulate transcription of a specific set of targets. This or closely related motifs are present in all seven *SMR* target loci at positions frequently coinciding with the peak summits of SCL28 and AtSMOS1 (Fig. 6c and Supplementary Fig. 16), suggesting that SCL28 and AtSMOS1 recognize this motif and bind to their targets as a heterodimer.

We then tested whether chromatin association of SCL28 depended on AtSMOS1 and vice-versa by conducting ChIP-qPCR experiments in *scl28* and *atsmos1* backgrounds (Fig. 6f). As expected, both SCL28 binding on *SMR2* promoter in the *atsmos1* and AtSMOS1 binding in the *scl28* mutant backgrounds were abolished. Collectively, our data supports SCL28 and AtSMOS1 functional heterodimer binding to specific sets of *SMR* target genes to activate transcription and negatively regulate cell cycle progression at the G2 to M cell cycle transition.

**SMRs are critical downstream effectors for SCL28 and AtSMOS1**. If *SMR* genes directly bound by SCL28 and AtSMOS1 are crucial for mediating their effects on cellular phenotype, transcript levels of those *SMR* genes should be altered accordingly with cell size phenotype observed in our epistasis analysis (see Fig. 4c–j). Confirming this idea, downregulation of target *SMR*

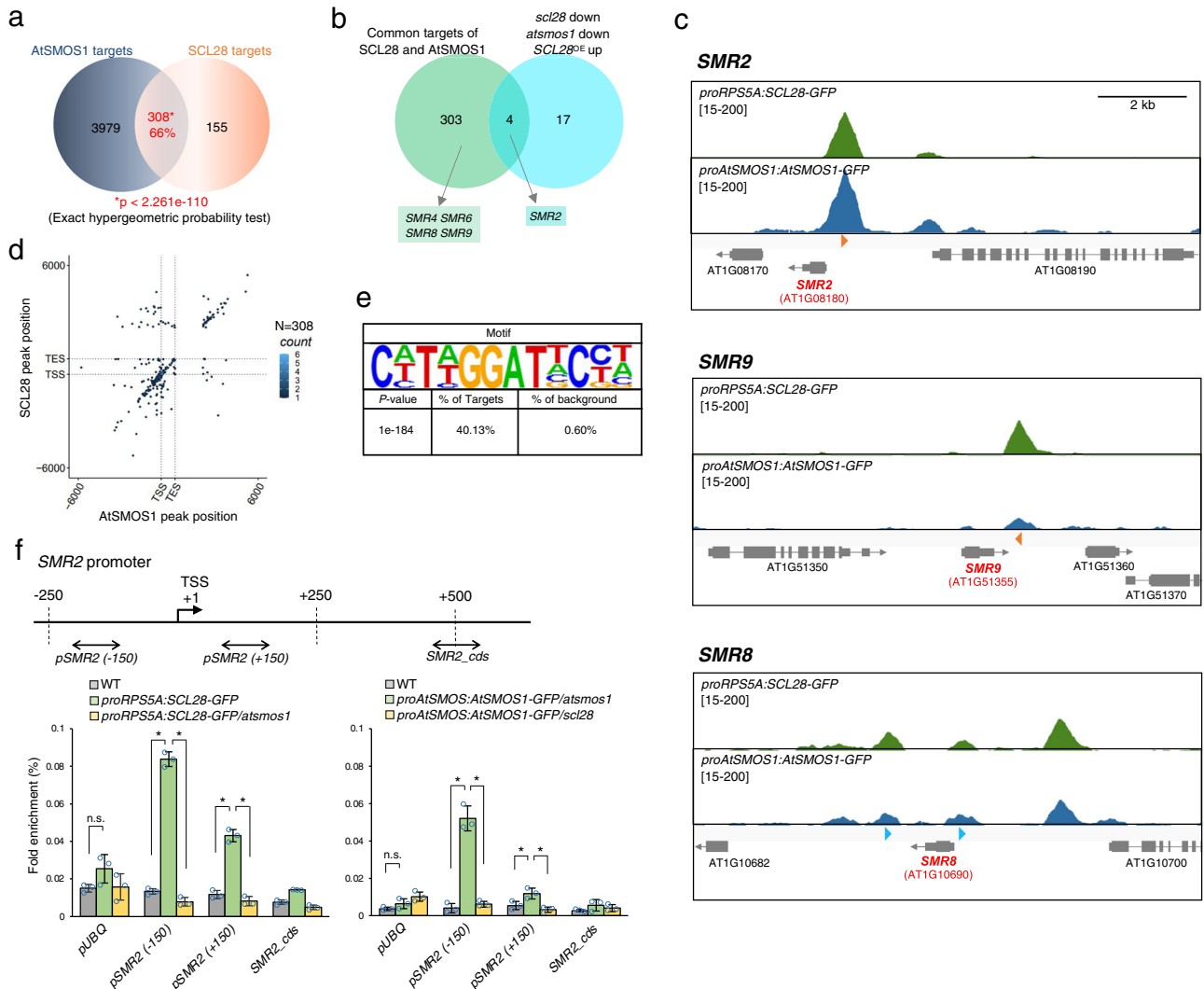

**Fig. 6 Binding of active SCL28-AtSMOS1 heterodimer to target *SMR* genes. a** Venn diagram showing significant overlap between genes bound by AtSMOS1 and SCL28. Significance of the overlap was evaluated using one-sided hypergeometric tests (no adjustment). **b** Venn diagram showing limited overlap between common target genes bound by SCL28 and AtSMOS1 and those downregulated in both *scl28* and *atsmos1* and upregulated in *SCL28*[OE]. *SMR* genes found in overlapping and non-overlapping regions are also shown. **c** ChIP-Seq profiles for SCL28 and AtSMOS1 around *SMR2*, *SMR8*, and *SMR9* loci. Orange arrowheads indicate DNA motifs perfectly matching C(a/t)T(a/t)GGATNC(c/t)(a/t), identified as an enriched motif in the SCL28/AtSMOS1 common targets, whereas blue arrowheads indicate motifs matching the enriched motif with one base mismatch (see Fig. 6e). **d** Density plot showing overlap of SCL28 and AtSMOS1 using hexagonal binning routine. Each dot represents the distance from the peak midpoint to the nearest gene. The *y*-axis shows location of the AtSMOS1 peak midpoint compared with gene position, while the *x*-axis indicates location of SCL28 peak midpoint compared with nearest gene. The large number of dots occurs along the positive correlation line, showing the co-occurrence pattern of SCL28 and AtSMOS1. **e** DNA motif overrepresented in common targets for SCL28 and AtSMOS1. HOMER motif search identified a specific common SCL28/AtSMOS1 binding motif with a *P* value of 1e−184 (one-sided hypergeometric tests, no adjustment), which was found in 40.13% of common targets. **f** ChIP-qPCR analysis was performed on *SMR2* loci using plants carrying proRPS5A::SCL28-GFP under WT and *atsmos1* backgrounds, and those carrying proAtSMOS1::AtSMOS1-GFP under *scl28* and *atsmos1* backgrounds. Amplified regions (−150, +150, and cds) in ChIP-qPCR are shown by double-headed arrows. *UBQ* locus was analyzed in the same way as a negative control without binding. Data are shown as averages from three technical replicates (±SD). Statistical significance was determined using two-sided Student's *t* test. **P* < 0.05. n.s., not significant.

genes—*SMR2*, *SMR9* and *SMR13*—were quantitatively equivalent in *scl28*, *atsmos1*, and double *scl28 atsmos1* mutants (Fig. 7a). In addition, strong activating effects of *SCL28*[OE] on target transcription was totally abolished under *atsmos1* mutant background, providing the molecular basis for AtSMOS1-dependent action of SCL28 on cell size (Fig. 7b). To further elucidate that AtSMOS1 acts together with SCL28 on *SMR* genes, we also analyzed the proSMR2::LUC reporter co-transfected in protoplasts with pro35::SCL28 and/or pro35S::AtSMOS1 constructs. The most prominent and significant activation of luciferase was

only observed when both plasmids expressing SCL28 and AtSMOS1 were simultaneously transfected, supporting the conclusion that SCL28 and AtSMOS1 fulfill their function as a transcriptional activator by forming a heterodimer (Fig. 7c).

To directly address whether these *SMR* genes are critical for SCL28 function, we performed genetic analysis focusing on *SMR1*, *SMR2* and *SMR13*, which are significantly downregulated in *scl28* and upregulated in *SCL28*[OE]. For each *smr* mutant, we could not detect any abnormalities in cell size (Supplementary Fig. 18). However, when these mutations were combined in a

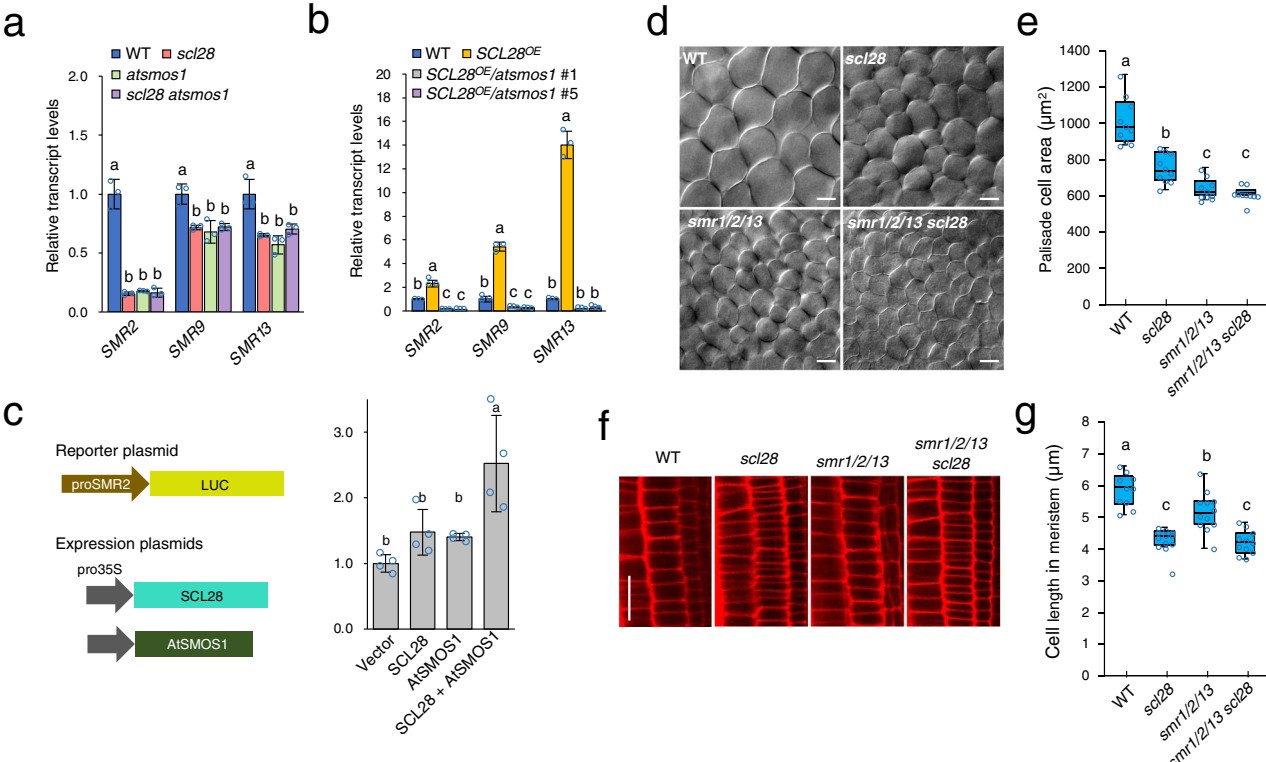

**Fig. 7 Cell size control by SCL28-AtSMOS1 is mediated by *SMR*s. a** Double *scl28 atsmos1* mutation downregulates *SMR* genes, equivalently to each single mutation. qRT-PCR analysis of *SMR2, SMR9* and *SMR13* transcripts was performed for comparing expression levels in WT, *scl28*, *atsmos1*, and *scl28 atsmos1* plants. Data are shown as averages from three biological replicates (±SD). Statistical analysis was performed for each *SMR* gene based on one-way ANOVA and Tukey's test, and significant differences (*P* < 0.05) were shown in different letters above the bars. **b** Upregulation of *SMR* genes in *SCL28*^OE requires presence of AtSMOS1. qRT-PCR analysis was performed for comparing expression levels between *atsmos1*, *SCL28*^OE plants, and those carrying *atsmos1* and *SCL28*^OE in combination. Two independent lines were used for analyzing *SCL28*^OE/ *atsmos1*. Data are shown as averages from three biological replicates (±SD). Statistical analysis was performed as in (**a**). **c** Co-expression of SCL28 and AtSMOS1 activates the *SMR2* promoter in the protoplast transient assay. proSMR2::LUC reporter plasmid was transfected into protoplasts prepared from T87 cells together with expression plasmids of SCL28 and AtSMOS1 as indicated. Schematic presentation of reporter and expression plasmids as shown on the left. Data are shown as averages from four biological replicates (± SD). Statistical analysis was performed as in (**a**). **d** DIC observations of leaf palisade cells in WT, *scl28*, *smr1/2/13*, and *smr1/2/13 scl28* plants at 22 DAS. Scale bar indicates 20 μm. **e** Quantification of palisade cell area in the first leaf pairs from plants with indicated genotypes. Boxplot was generated using data collected from leaves of ten different plants, in each of which more than 60 cells were analyzed (midline = median, box = IQR, whiskers = 1.5 × IQR). Different letters above the boxplots indicate significant differences (*P* < 0.05) based on one-way ANOVA and Tukey's test. **f** CLSM observation of cortical cell files in root meristems from WT, *scl28*, *smr1/2/3*, and *smr1/2/13 scl28* plants at 7 DAS. Scale bar indicates 20 μm. **g** Quantification of cortical cell length in root meristems from plants with indicated genotypes. Boxplot was generated using data collected from roots of multiple individual plants (midline = median, box = IQR, whiskers = 1.5 × IQR). Numbers of plants analyzed were 10, 12, 12, and 11 for WT, *scl28*, *smr1/2/3*, and *smr1/2/13 scl28*, respectively. In each plant, more than 40 cells were analyzed for calculating average, which were further used for statistical analysis. Different letters above boxplots indicate significant differences as in (**e**).

*smr1/2/13* triple mutant, there was a significant reduction in cell size in multiple tissues, including leaf palisades (Fig. 7d, e) and root meristem (Fig. 7f, g), suggesting functional redundancy among *SMR* genes. To address the link between SCL28 and SMRs, we studied the cell size phenotype when these mutations were combined. In leaf palisade cells, the triple *smr1/2/13* mutant showed stronger reduction in cell size than the *scl28* mutant, which did not further reduce when all mutations combined in the *smr1/2/13 scl28* line, supporting the idea that cell size regulation by SCL28 is mediated by SMR1/2/13 (Fig. 7d, e). Conversely, *scl28* mutation still significantly decreased cell size in the root meristem when combined with *smr1/2/13* (Fig. 7f, g). Therefore, SMR1/2/13 contribution downstream of SCL28 appears to be larger in palisade tissue than root meristem, where additional SCL28-regulated *SMRs* may play a role in cell size control. These tissue-specific differences indicate the contribution of developmental regulation that positions different sets of *SMRs* downstream of SCL28.

**Dose-dependent control of cell size and cell number by SCL28 to maintain organ size homeostasis.** In the *scl28* mutant, leaf growth is essentially normal, but the constituent cells became small and more numerous (see Fig. 2d, e, g), suggesting that SCL28 is pivotal to maintain organ size homeostasis by regulating the balance between cell size and cell number. For a regulator designed to tune the balance between cellular parameters, one assumes that it acts dose dependently. To explore whether or not SCL28 levels quantitatively affect cell size, we analyzed plants heterozygous for *scl28* (*scl28*/+ plants). Heterozygous plants showed reduced *SCL28* transcript levels that is approximately half of that in WT (Fig. 8a). This *SCL28* downregulation resulted in a clear reduction of cell size in root meristem and leaf palisade tissue, with an intermediate cell size in heterozygous plants between those in WT and *scl28* homozygous plants (Fig. 8b, c and Supplementary Fig. 19a, b).

Overexpression of *SCL28* causes a phenotype opposite to *scl28*, dramatically reducing the number of cells that became

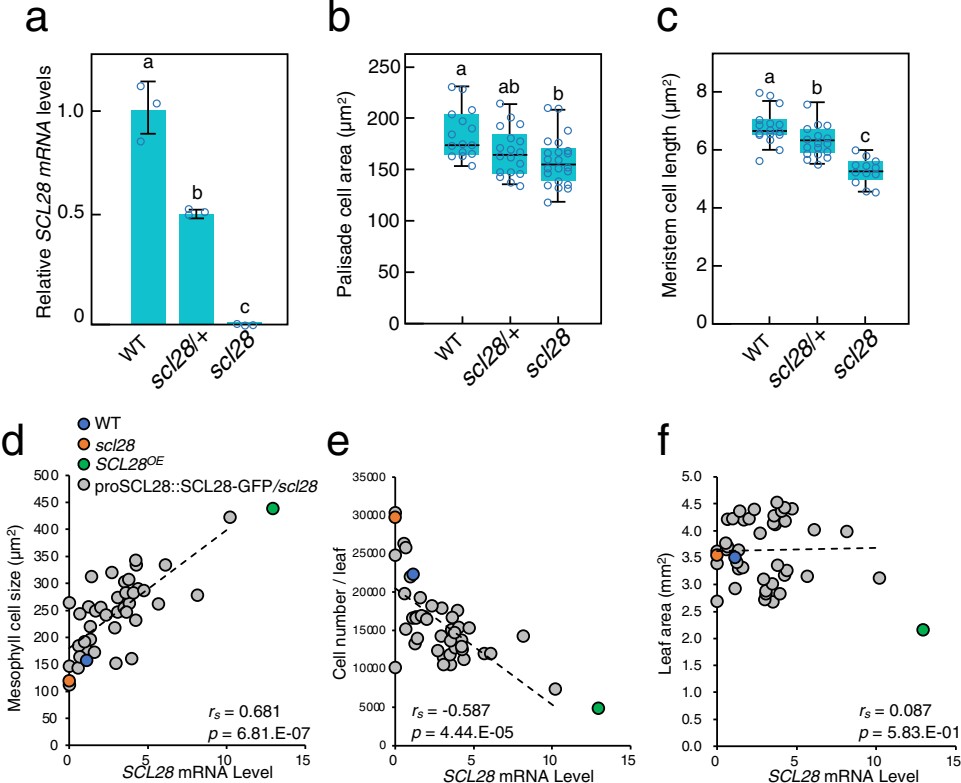

**Fig. 8 SCL28 regulates cell size in a dose-dependent manner. a** Moderate downregulation of *SCL28* transcripts in plants heterozygous for *scl28*. qRT-PCR analysis was performed using WT plants, and those heterozygous (*scl28/+*) and homozygous (*scl28*) for *scl28* at 10 DAS. Data are shown as averages from three biological replicates (±SD). Statistical analysis was performed based on one-way ANOVA and Tukey's test and significant differences ($P < 0.05$) are shown in different letters above bars. **b** Quantification of palisade cell area in *scl28*, *scl28/+* and WT plants. Boxplot was generated using cell size data from 1st leaf pairs of multiple individual plants at 10 DAS (midline = median, box = IQR, whiskers = 1.5 × IQR). Numbers of plants analyzed were 15, 18, and 22 for *scl28*, *scl28/+* and WT, respectively. In each plant, more than 70 cells were analyzed for calculating average, which were further used for statistical analysis. Different letters above boxplots indicate significant differences ($P < 0.05$) based on one-way ANOVA and Tukey's test. **c** Quantification of cortical cell length in root meristems of *scl28*, *scl28/+* and WT plants. Boxplot was generated using cell size data from roots of multiple individual plants at 7 DAS (midline = median, box = IQR, whiskers = 1.5 × IQR). Numbers of plants analyzed were 16, 17, and 12 for *scl28*, *scl28/+* and WT, respectively. In each plant, more than 30 cells were analyzed for calculating average, which was further used for statistical analysis. Different letters above the boxplots indicate significant differences as in (**b**). **d** Correlation between *SCL28* mRNA levels and palisade cell size analyzed by scatterplots, where each dot corresponds to each individual plant from different transgenic lines carrying proSCL28::SCL28-GFP under *scl28* background. In addition to proSCL28::SCL28-GFP plants (shown in gray dots), WT (blue dot), *scl28* (red dot), and *SCL28*[OE] plants (green dot) were also analyzed. **e** Correlation between *SCL28* mRNA levels and number of palisade cells per leaf was analyzed by scatterplot as in (**d**). Symbols are same as in (**d**). **f** Correlation between *SCL28* mRNA levels and leaf area was analyzed by scatterplot as in (**d**). Symbols are same as in (**d**). Values in (**d–f**) represent Pearson correlation coefficients ($r_s$) and its two-sided $P$ value without adjustment ($p$).

enlarged. To examine the outcome when SCL28 expression is only moderately increased, we utilized plants from different T2 lines carrying proSCL28::SCL28-GFP in *scl28* mutant background and showing varying levels of SCL28-GFP expression. We quantitatively compared levels of *SCL28* transcript and cellular parameters of palisade tissues in each individual plant (Fig. 8d, e and Supplementary Fig. 19c). This showed that SCL28 expression positively correlates with cell size and negatively with leaf cell number. Both positive and negative correlations were statistically significant. Due to compensatory changes in cell number and size, overall leaf size was not dramatically affected by moderate alteration of SCL28 expression (Fig. 8f). Therefore, we concluded that a finely-graded expression of SCL28 sets the balance between cell size and number without altering organ growth as a whole. This mechanism may enable plants to achieve an optimal balance between cell size and number, possibly depending on developmental status and environmental conditions.

## Discussion

Cell size at division depends on coordination between cell growth and cell cycle progression, which is actively maintained in a cell autonomous manner in both uni- and multicellular organisms[36–38]. Here, we identified a molecular mechanism for cell size regulation in Arabidopsis that relies on a hierarchical transcriptional activation of CDK inhibitors. In this pathway, SCL28 expression is specifically confined to mitosis through the MSA element in its promoter controlled by MYB3Rs. In turn, SCL28 associates with AtSMOS1, a transcription factor with cell cycle-independent expression, and this dimer defines the binding site present in a set of *SMR* genes encoding CDK inhibitors to activate their expression and thereby act as a brake against the cell cycle engine predominantly driven by CDK activity (Fig. 9). The SCL28-AtSMOS1-SMR axis uniquely affects only cell cycle progression and cell doubling time, but not the exit from cell proliferation. Therefore, it acts to set the balance between cell number and size during organ development without significantly

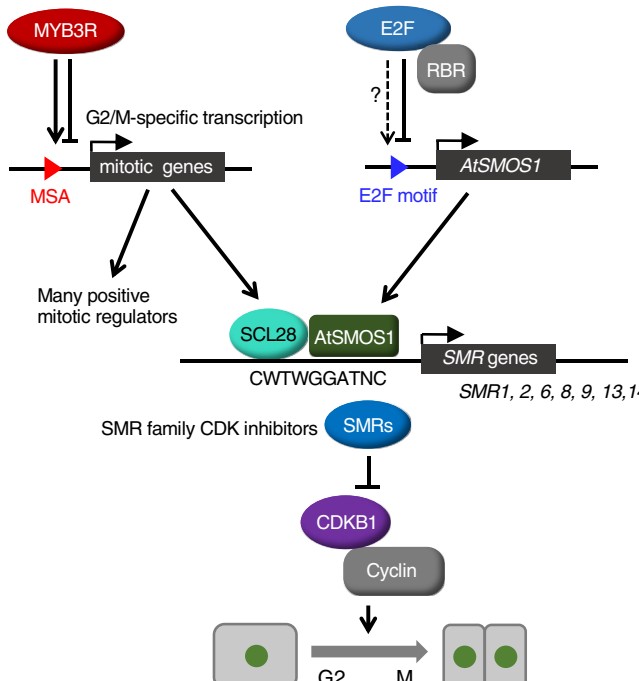

**Fig. 9 Schematic model of SCL28-dependent cell size control.** MYB3R transcription factors regulate a suite of G2/M-specific genes, including many positive mitotic regulators. However, *SCL28*, similarly regulated by MYB3Rs acts as a negative regulator of the cell cycle together with *AtSMOS1*, which may be under the control of the E2F-RBR pathway. Formation of the SCL28-AtSMOS1 heterodimer generates specific binding sequences, enabling direct transcriptional activation of *SMR* family genes. An increasing amount of SMR proteins, in turn, inhibits CDK activity, thereby negatively influencing cell cycle progression during G2. By controlling the activity of SCL28-AtSMOS1 heterodimer, G2 duration in the cell cycle can be finely tuned to maintain a proper balance between cell size and cell number in developing plant organs.

impacting final organ size. Whether SCL28 is part of a cell size sensor and how it is coupled to the measurement of cell growth remain to be determined.

We provide multiple lines of evidence supporting the conclusion that SCL28 acts as a brake on the G2/M transition to set cell size: (1) measurements of cell cycle phases by following PCNA-GFP using live-cell imaging showed that *scl28* mutant and *SCL28*[OE] shortened or lengthened the G2 phase, respectively; (2) kinematic analysis of cellular parameters during leaf development of *scl28* and *SCL28*[OE] showed a clear alteration in cell doubling time without any effect on cell cycle exit; (3) the molecular mechanisms identified here are fully consistent with the cell cycle inhibitory function, SCL28, forming a dimer with AtSMOS1 and directly activating the transcription of *SMR* genes whose products inhibit CDK activity; (4) in support of this, the *scl28* mutant phenocopied and interacted in an epistatic fashion with the triple *smr1/2/13* mutant; and 5) SCL28 regulates the number of endoreplication cycles, which is typically enhanced by G2 inhibition. Previous reports suggest that SCL28 plays a positive role in root growth and promotes the G2/M cell cycle progression[23,39]. These conclusions are inconsistent with our data and the small cell size phenotype observed in both studies, which may originate from using different methods relying on fixed samples.

Another example of a transcription factor inhibiting cell cycle at G2/M is a NAC-type transcription factor, SUPPRESSOR OF GAMMA RESPONSE 1 (SOG1), acting as a central regulator of DNA damage response in Arabidopsis. Similar to SCL28, SOG1

transcriptionally activates *SMR* family genes to inhibit cell cycle progression upon DNA damage[40], but its targets, *SMR5* and *SMR7*, are not regulated by SCL28, which instead directly regulates seven other *SMR* genes. Notably, in contrast with SOG1, which is activated specifically upon DNA damage, SCL28 negatively regulates cell cycle, thereby affecting cell size during normal plant development. The target of rapamycin (TOR) signaling pathway is a primary regulator of cell growth and proliferation and is involved in cell size regulation in yeast[41]. In Arabidopsis, TOR, acting through YAK1 kinase, regulates the transcription of *SMR* genes and thereby G2/M transition[42]. However, the set of YAK1-targeted *SMRs* overlap with those regulated by SOG1 rather than those regulated by SCL28. Therefore, our study places SCL28 as part of a novel pathway for regulating G2/M progression that determines cell size and cell number during plant organ development.

Variability in cell size at birth or stochastic changes in key regulators necessitate mechanisms that achieve cell size homeostasis by linking cell cycle progression to cell growth[38]. The way cell size variations are corrected has long been debated. The principal possibilities are using (i) a cell cycle timer, (ii) cell size measurements or (iii) correcting cell size by adding a constant increase to daughters regardless of initial sizes[36]. Comparing predicted outcomes of these theoretical models with time-lapse imaging data on cell size in plant meristematic cells identifies possible co-existence of a mix of these mechanisms that may act at multiple control points of the cell cycle[31,43]. The molecular device for cell size control may rely on cell size-dependent accumulation of an activator or dilution of a cell cycle inhibitor molecule. The 'inhibitor dilution model' postulates that cell growth dilutes such inhibitor protein until its concentration meets a threshold, thereby permitting cell division when cells reach an appropriate size[44]. This hypothesis has been verified in yeast and mammalian systems[45,46]. In Arabidopsis, recent studies show segregation of set amounts of chromatin-bound KRP4 CDK inhibitors act as a cell size sensor that is diluted by cell growth and is responsible for cell size-dependent cell cycle regulation at G1/S[47].

Live-cell imaging of cell cycle markers in Arabidopsis shoot apical meristem demonstrated that both G1/S and G2/M transitions are regulated in a cell size-dependent manner. Accumulation of two waves of CDK activities underpinned by D-type cyclin and mitotic CDKB1, their interactors or downstream effectors were proposed as molecular components for cell size-dependent G1/S and G2/M regulation, respectively[31]. SCL28 may participate in regulating G2/M transition in this model, in which dilution of an inhibitor rather than accumulation of an activator is the principle behind. How SCL28 is connected to monitoring of cell size remains to be determined. Stochastic variability in the accumulation of key regulators can also set cell size and cell fate differences in a developing organ. An example is ATML1, a transcription factor that is necessary for the patterning of giant cells within the sepal epidermis by overcoming a threshold in G2 cells and activating the downstream *SMR1*, also called LOSS OF GIANT CELLS FROM ORGANS (LGO)[48]. Similar to ATML1, SCL28 shows a finely-tuned dose-dependent effect on cell size by accurately distinguishing gene copy numbers of *SCL28* and reflecting its graded overexpression. It is possible that the molecular ruler in an inhibitor dilution model potentially involving SCL28 is set by the synthesis rate limited by gene copy number, as shown for Whi5 in budding yeast[45].

The human functional orthologue of Whi5 is pRB, which also controls cell size via its synthesis mostly after G1 and subsequent dilution during growth[46]. In Arabidopsis, both RBR and E2FB were shown to strongly affect cell size in proliferating cells[49,50], but as opposed to SCL28 that solely acts on cell proliferation rate,

the RBR-E2F pathway also regulates cell cycle exit, establishment of quiescence, and cellular differentiation[29,51,52]. The repressor-type MYB3Rs—MYB3R1, MYB3R3 and MYB3R5—when combined with RBR and E2FC form the DREAM complex that represses mitotic genes during or after developmental cessation of cell proliferation[11]. Whether the RBR/E2F/DREAM pathway directly regulates the cell size in proliferating cells or its cell size effect is through the regulation of cell cycle exit, remains to be established. It will be also important to know how the RBR/E2F/DREAM and the SCL28/AtSMOS1 pathways are coordinated.

In addition to cell size during proliferation, SCL28 also strongly affects the size of terminally differentiated cells. In the scl28 mutant, we observed a reduced final cell size in various organs, whereas moderately higher SCL28 expression led to its increase. Reduced cell size caused by increased cell division, as observed in scl28, has been repeatedly reported in Arabidopsis organ development. Examples include plants with inhibited RBR expression, those overexpressing CYCD3;1, or co-overexpressing E2FA and DPA[50–53]. In many of these cases, the cellular phenotype is due to prolonged cell proliferation and delayed cell differentiation, leading to additional cell division that is not balanced by cell size increase due to developmental decrease of cell expansion. On the contrary, we showed with kinematic analysis during leaf development that the increased number of smaller cells observed in scl28 leaves is caused by accelerated cell proliferation, but not by prolonged cell proliferation. Therefore, it is puzzling how loss of SCL28 affects final cell size that is largely determined by post-mitotic cell expansion occurring in the absence of SCL28.

Our observations are reminiscent of the phenomena reported in Drosophila wings, where the alteration of cell number is compensated by changes in cell size such that final organ size is essentially unchanged. Based on this and similar observations, it has been proposed that animal organ size may be controlled by 'total mass checkpoint' mechanisms, which operate at the level of whole organs[54], although the underlying molecular details remain unknown. Instead, the well-established mechanism of 'compensation' in plants shows post-mitotic cell expansion that is enhanced by reduced cell proliferation[55]. A small number of larger cells observed in SCL28[OE] leaves may be a prime example of compensation. One possible explanation for compensatory alteration of cell expansion is that, during cell proliferation, cells establish memory that is maintained during the elongation phase. An example of this can be seen in plants overexpressing KRP2, where final cell size may be influenced by the size of mitotic cells that are already altered during proliferation[56]. We observed larger and smaller mitotic cells both in root meristem and young leaves of SCL28[OE] and scl28, respectively, which may be maintained until terminal differentiation, thus affecting final cell size. A possible molecular mechanism to create such memory is cell size-dependent alteration in gene expression. It was shown in yeast that cell size changes due to ploidy levels are associated with altered gene expression related to functions of cell wall and extracellular matrix[57]. Similarly, in Arabidopsis root, ploidy levels were correlated with expression of genes related to chromatin and cell expansion, such as ion transport and cell wall modifications[58]. Therefore, it is plausible that SCL28 action in meristematic cells has long-lasting effects through cell expansion on final cell size in differentiated tissues. Another possible mechanism for cell size memory involves a monitoring system for cellular dimensions that shares components in proliferating and expanding cells. Supporting this idea, it has been suggested that cells expand to a target size by a mechanism requiring a cell size measurement device, from an approach combining quantitative analysis and mathematical modeling of variability in final cell size of Arabidopsis root[59]. Further careful studies are required to clarify how final cell size is affected by expression levels of SLC28 that is confined to proliferating cells.

In summary, we identified a novel hierarchical MYB3R-SCL28/AtSMOS1-SMR transcriptional regulatory pathway leading to cell cycle inhibition at G2/M that regulates cell size and number during organ development without dramatically altering organ size. This mechanism may help to adjust cell size to optimize cellular and tissue performance such as metabolic functions or mechanical properties.

## Methods

**Plant materials**. *Arabidopsis thaliana* Columbia (Col) was used as the WT plant. All mutants and transgenic lines used in this study were in a Col background. The mutant alleles scl28-1 (SALK_205284), atsmos1-1 (SALK_111105C), smr1-1 (SALK_033905), smr2-1 (SALK_006098C), and smr13-1 (SALK_053256C) were identified from SALK T-DNA collection, and used for phenotype analysis and generating multiple mutant combinations. Other mutants and transgenic lines, namely myb3r1-1, myb3r3-1, myb3r4-1, myb3r5-1, PCNA-RFP, and CYCB1;1-GFP, were described previously[11,13,28,30]. Sterilized seeds were germinated on one-half-strength Murashige and Skoog (1/2 MS) medium containing 2% sucrose and 0.6% agar. Plants were grown on 1/2 MS agar medium or soil under continuous light at 22 °C. For root phenotype analysis, plants were gown on a vertical surface of 1/2 MS medium containing 1.0% agar.

**qRT-PCR analysis**. Total RNA was extracted using TRIzol reagent (Thermo Fisher) and used for synthesis of first-strand cDNA with ReverTra Ace qPCR RT Master Mix (Toyobo), according to the manufacture's instruction. A real-time qPCR was conducted using Thunderbird SYBR qPCR Mix (Toyobo) on StepO-nePlus Real-Time PCR Systems (Applied Biosystems). Three biological replicates were run for each measurement, and results were normalized to the expression levels of UBQ5 mRNA unless otherwise mentioned. Primers used for qRT-PCR are listed in Supplementary Table 1.

**Yeast two-hybrid assay**. For yeast two-hybrid assays, *Saccharomyces cerevisiae* strain L40 [MATα his3-200 trp1-901 leu2-3,112 ade2 LYS2::(lexAop)4-HIS3 URA3::(lexAop)8-lacZ] was transformed with pBTM116- and pVP16-based plasmids[60] carrying coding sequences encoding for SCL28 or AtSMOS1 using S.c. EasyComp Transformation Kit (Thermo Fisher). Transformants were selected on the synthetic medium lacking Ura, Leu and Trp at 30 °C for 2–3 days. A single colony was diluted with water and spotted on synthetic medium lacking Ura, Leu, Trp and His supplemented with 3-AT (50 or 100 mM). After incubation at 30 °C for 2 days, the yeast transformants grown on the medium were photographed.

**BiFC assay**. For construction of plasmids used for BiFC assays, the entire coding regions of SCL28 and AtSMOS1 were amplified by PCR, and cloned into donor vectors pDONR201 and pDONR207 using BP Clonase II (Thermo Fisher), respectively. The cloned fragments were then transferred into the destination vectors pGWnY and pGWcY[61] using LR clonase II (Thermo Fisher) to generate C-terminal fusions of YFP fragments. Transient gene expression using Arabidopsis leaf mesophyll protoplasts was performed as described previously[62]. The transfected protoplasts were incubated overnight in the dark at 22 °C. YFP fluorescent was observed with fluorescent optics on a BX54 microscope (Olympus).

**LUC reporter assay**. Preparation of protoplasts from Arabidopsis T87 suspension cultured cells and polyethylene glycol-mediated gene transfer were performed as described previously[63]. An empty vector containing 35 S promoter (pJIT-60) and 35 S:hRLUC plasmid expressing the humanized *Renilla* LUC (hRLUC) were used as a negative control and an internal control, respectively. Protoplasts ($1.5 \times 10^5$ for each transfection) were co-transfected with 15 μg each of the LUC reporter and expression plasmids, then incubated at 22 °C for 20 h before measuring LUC activities, which was performed using the Dual-Luciferase Reporter Assay system (Promega) and Luminoskan Ascent luminometer (Thermo Fisher). LUC activity was normalized according to hRLUC activity in each assay, and the relative ratio was determined.

**Plasmid construction**. To construct GUS fusion reporters, the upstream region of SCL28 (2.2 kb) was amplified by PCR and cloned into the pENTR/D-TOPO vector (Thermo Fisher), then transferred to pBGGUS[10] through Gateway LR reaction to create proSCL28::GUS. The PCR-based site-directed mutagenesis was performed to change all four MSA core motifs, AACGG, in the SCL28 promoter to ATTGG, resulting in generation of proSCL28ΔMSA::GUS.

To prepare the construct expressing SCL28 fused to GFP at its C-terminus (SCL28-GFP) under the control of the native promoter, the entire SCL28 genomic region (5.3 kb) containing 2.2-kb promoter was amplified by PCR using genomic DNA from Arabidopsis (Col), and cloned into pENTR/D-TOPO. The resulting plasmid was then used for In-Fusion reaction (Takara Clontech) to insert PCR-

amplified GFP fragments at the C-terminus of the *SCL28* coding sequence, to create entry plasmid containing the proSCL28::SCL28-GFP fusion construct, which was then transferred to the binary vector pGWB501[64].

To prepare the AtSMOS1-GFP fusion construct driven by its own promoter, the entire *AtSMOS1* genomic region (4.4 kb) containing 1.2 kb promoter was amplified by PCR using genomic DNA from Arabidopsis (Col) and cloned into pDONR201 through Gateway BP reaction (Thermo Fisher). The resulting plasmid was then used for In-Fusion reaction (Takara Clontech) for inserting PCR-amplified GFP fragment at the C-terminus of *AtSMOS1*, to create entry plasmid containing proAtSMOS1::AtSMOS1-GFP fusion construct, which was then transferred to the binary vector pPZP211-GW[65].

For construction of a LUC reporter plasmid, promoter region of *SMR2* (2.0 kb) was amplified by PCR and cloned into *Hin*dIII-*Bam*HI interval of pUC-LUC[12] to obtain proSMR2::LUC. For construction of expression plasmids of AtSMOS1 and SCL28 (pro35S::AtSMOS1 and pro35S::SCL28), the entire coding regions were amplified by PCR using cDNA prepared from Arabidopsis T87 cells as a template. The resulting PCR fragments were cloned into pENTR/D-TOPO and then transferred to pJIT60[12] at the site downstream of 35S promoter through the Gateway LR reaction.

To create proRPS5A::SCL28-GFP, the entire coding sequence of SCL28-GFP fusion was amplified by PCR using proSCL28::SCL28-GFP plasmid as a template and cloned into pENTR/D-TOPO. The resulting plasmid was then used for the Gateway LR reaction to transfer the SCL28-GFP fragment downstream of *RPS5A* promoter in pPZP221[13]. Primers used for plasmid construction are listed in Supplementary Table 1.

**Histological analysis**. Excised plant organs were fixed in FAA solution (100% ethanol:formaldehyde:glacial acetic acid:water = 20:19:1:1) under vacuum, stained with 1% Safranin O, dehydrated with 30%, 50%, 70%, and 100% ethanol series, and embedded in Technovit 7100 (Heraeus Kulzer). 2 μm-thickness sections were cut on RM2125RT microtome (Leica) equipped with a tungsten carbide disposable blade TC-65 (Leica). Sections were briefly stained in a 0.01–0.5% toluidine blue-O in 0.1% Na$_2$CO$_3$ solution, then washed with 0.1% Na$_2$CO$_3$ solution. Sections were observed under BX63 microscope (Olympus) and images were acquired with cellSens Standard Software (Olympus).

**Transcriptome analysis**. Microarray analyses were performed using an ATH1 GeneChip (Affymetrix). Total RNA was extracted from whole seedlings by RNeasy Plant Mini Kit (Qiagen), and reverse-transcribed, labeled with an Affymetrix 3′ IVT Express Kit (Affymetrix), and used for hybridization to the chip according to the supplier's protocol. Data analysis was performed using Microarray Suite ver. 5 (Affymetrix) and GeneSpring 7.1 (Agilent Technologies). For transcriptome profiling in WT and *SCL28*$^{OE}$ plants, whole seedlings at 9 DAS were analyzed with three biological replicates. Genes with FDR < 0.05 were defined as differentially expressed genes in *SCL28*$^{OE}$ compared with WT plants.

For RNA-Seq analyses, total RNA was extracted as above and used to construct cDNA libraries with the TruSeq RNA Library Preparation Kit v2 (Illumina, United States) according to the manufacturer's protocol. For transcriptome profiling in WT, *scl28*, and *atsmos1* plants, whole seedlings at 9 DAS were analyzed with three biological replicates. The libraries were sequenced using the NextSeq500 sequencer (Illumina, United States). Raw reads containing adapter sequences were trimmed using bcl2fastq (Illumina, United States), and nucleotides with low-quality (QV < 25) were masked by N using the original script. Reads shorter than 50 bp were discarded, and the remaining reads were mapped to the cDNA reference using Bowtie with the following parameters: "–all–best–strata"[66]. The reads were counted by transcript models. Differentially expressed genes were selected based on the adjusted *P*-value calculated using edgeR (version 3.20.9) with default settings[67].

**ChIP-Seq assay**. ChIP-Seq assays were performed on whole seedlings using anti-GFP antibody (Abcam, ab290). Seedlings at 14 DAS (5 g) from proAtSMOS1::AtSMOS1-GFP and proSCL28::SCL28-GFP were crosslinked in 1% (v/v) formaldehyde at room temperature for 15 min. Crosslinking was then quenched with 0.125 M glycine for 5 min. The crosslinked seedlings were ground, and nuclei were isolated and lysed in nuclei lysis buffer (1% SDS, 50 mM Tris-HCl, 10 mM EDTA, pH 8.0). Crosslinked chromatin was sonicated using a water bath Bioruptor UCD-200 (Diagenode) (15 s on/15 s off pulses; 15 times). The complexes were immunoprecipitated with antibodies (1 μg for each immunoprecipitation), overnight at 4 °C with gentle shaking, and incubated for 1 h at 4 °C with 40 μL of Protein AG UltraLink Resin (Thermo Fisher). The beads were washed 2 × 5 min in ChIP Wash Buffer 1 (0.1% SDS, 1% Triton X-100, 20 mM Tris-HCl, 2 mM EDTA, 150 mM NaCl, pH 8.0), 2 × 5 min in ChIP Wash Buffer 2 (0.1% SDS, 1% Triton X-100, 20 mM Tris-HCl, 2 mM EDTA, 500 mM NaCl, pH 8.0), 2 × 5 min in ChIP Wash Buffer 3 (0.25 M LiCl, 1% NP-40, 1% sodium deoxycholate, 10 mM Tris-HCl, 1 mM EDTA, pH 8.0) and twice in TE (10 mM Tris-HCl, 1 mM EDTA, pH 8.0). ChIPed DNA was eluted by two 15 min incubations at 65 °C with 250 μL elution buffer (1% SDS, 0.1 M NaHCO$_3$). Chromatin was reverse-crosslinked by adding 20 μL of 5 M NaCl and incubated overnight at 65 °C. Reverse-crosslinked DNA was subjected to RNase and proteinase K digestion and extracted with phenol-chloroform. DNA was ethanol precipitated in the presence of 20 μg of glycogen and

resuspended in 50 μL of nuclease-free water (Ambion) in a DNA low-bind tube. We used 10 ng of IP or input DNA for ChIP-Seq library construction using NEBNext® Ultra DNA Library Prep Kit for Illumina® (New England Biolabs) according to manufacturer's recommendations. For all libraries, twelve cycles of PCR were used. Library quality was assessed with Agilent 2100 Bioanalyzer (Agilent).

**Computational analysis of ChIP-Seq data**. Single-end sequencing of ChIP samples was performed using Illumina NextSeq 500 with a read length of 76 bp. Reads were quality controlled using FASTQC (http://www.bioinformatics.babraham.ac.uk/projects/fastqc/). Trimmomatic was used for quality trimming. Parameters for read quality filtering were set as follows: minimum length of 36 bp; mean Phred quality score greater than 30; and leading and trailing bases removal with base quality below 5. Reads were mapped onto the TAIR10 assembly using Bowtie[68] with mismatch permission of 1 bp. To identify significantly enriched regions, we used MACS2[69]. Parameters for peak detection were set as follows: number of duplicate reads at a location: 1; mfold of 5: 50; *q* value cutoff: 0.05; extsize 200; and sharp peak. Visualization and analysis of genome-wide enrichment profiles were performed with Integrated Genome Browser. Peak annotations such as proximity to genes and overlap of genomic features, including transposons and genes were performed using BEDTOOLS INTERSECT. NGSplot was used to profile enrichment at TSSs and along the gene[70]. Spatial binding of the AtSMOS1 and SCL28 peaks were performed by position-wise comparison using a binning approach and plotted in hexplot. De novo motif analysis of both SCL28 and AtSMOS1 binding regions were screened using HOMER[71].

**ChIP-qPCR analysis**. Whole seedlings were cross-linked in 20 mL of 1% formaldehyde solution under vacuum for 30 min. Plants were then washed twice with 0.125 M glycine solution, ground to powder in liquid nitrogen and resuspended in 15 mL Extraction buffer I (0.4 M sucrose, 10 mM Tris-HCl, pH 8.0, 10 mM MgCl$_2$, and complete protease inhibitor cocktail tablets [Roche]). After filtration with Miracloth (Millipore), samples were centrifuged and resulting pellets were resuspended in 1 mL of Extraction buffer II (0.25 M sucrose, 10 mM Tris-HCl, pH 8.0, 10 mM MgCl$_2$, and complete protease inhibitor cocktail tablets [Roche]). After centrifugation, the pellets were resuspended in 300 μL of Extraction buffer III (1.7 M sucrose, 10 mM Tris-HCl, 2 mM MgCl$_2$, 0.15% Triton-X-100, and complete protease inhibitor cocktail tablets [Roche]) and then loaded on equal volume of Extraction buffer III and centrifuged. The pellets were dissolved in Lysis buffer (50 mM Tris-HCl, 10 mM EDTA, 1.0% SDS, and complete protease inhibitor cocktail tablets [Roche]), sonicated with BIORAPTOR II ultrasonic disruptor (CosmoBio) and centrifuged. The supernatant was diluted 10 times with ChIP dilution buffer (16.7 mM Tris-HCl, pH 8.0, 1.2 mM EDTA, 167 mM NaCl and 1.1 % Triton-X-100) and used for immunoprecipitation. Immunoprecipitation of chromatin complexes was performed with anti-GFP antibody (ab290; Abcam, 5 μg for each immunoprecipitation), which was bound to Dynabeads Protein G (Invitrogen) for 1 h at 4 °C. Beads were washed twice with low salt buffer (20 mM Tris-HCl, pH 8.0, 2 mM EDTA, 150 mM NaCl, 0.5% Triton-X-100 and 0.2% SDS), once with high salt buffer (20 mM Tris-HCl, pH 8.0, 2 mM EDTA, 400 mM NaCl, 0.5% Triton-X-100 and 0.2% SDS) and once with LiCL buffer (10 mM Tris-HCl, 1 mM EDTA, 250 mM LiCl, 1% NP-40 and 1% sodium deoxycholate). Chromatin was then eluted from the beads using elution buffer (50 mM Tris-HCl, pH 8.0, 10 mM EDTA, 100 mM NaCl, and 1% SDS). Cross-linking of chromatin was reversed by incubating at 65 °C for 12 h followed by digestion with Proteinase-K (#9033; Takara). DNA was purified by NucleoSpin (Macherey-Nagel) according to the manufacturer's instruction and used for quantitative PCR with specific primer pairs (Supplementary Table 1).

**Kinematic analysis of leaf growth**. After plants were fixed in a 9:1 of ethanol and acetic acid solution and cleared with Hoyer's solution (a mixture of 100 g chloral hydrate, 10 g glycerol, 15 g gum arabic, and 25 mL water), we performed microscopic observations using 1st or 2nd leaves as described previously[13]. After whole leaf images were captured, palisade cells at positions one-fourth and three-fourth from the tip of each leaf were observed with differential interference contrast (DIC) microscope (BX51, Olympus). The captured images were analyzed using ImageJ (ver.2.1.0; rsb.info.nih.gov/ij) and the average size of palisade cell, the number of cells in the uppermost layer of palisade tissue per leaf, and cell division rate were calculated according to methods described previously[72].

**Ploidy analysis**. For ploidy analysis, whole leaves from plants at 8–20 DAS were used. Nuclei were isolated by chopping whole leaves with razor blade in Nuclei extraction buffer of CyStain UV precise P kit (Sysmex), filtered through 30 μm mesh, and stained with DAPI by adding Staining buffer of CyStain UV precise P kit. After incubation for 10 min, samples were analyzed with CyFlow Ploidy Analyzer (Sysmex) according to the manufacturer's instructions. For estimating population of nuclei in each peak, baselines of ploidy distribution profiles were calculated using the polynomial trendline function of Microsoft Excel. The value of baseline at each position was subtracted from the corresponding value of raw data for calculating the total count in each peak.

**Analysis of meristem cell number and cell size in roots**. To visualize cell outlines, roots from plants at five DAS were stained with 0.05 mg mL$^{-1}$ propidium iodide and observed by confocal laser scanning microscopy (CLSM), using an inverted fluorescence microscope (Eclipse Ti2, Nikon) equipped with a confocal scanning unit (A1, Nikon). The resulting images were processed using ImageJ software to measure cell length in the root meristem. Root meristem size was measured by counting the number of cortical cells between the quiescent center and the first elongated cell.

**Cell cycle analysis by time-lapse imaging**. Seedlings at six DAS carrying PCNA-GFP were transferred onto an MS medium in a glass-bottom dish, and root meristem cells were observed by CLSM as described above. Time-lapse images were acquired every 30 min for 15 h. To avoid long-time imaging that possibly damages the samples, the duration of the G1/S and G2/M phases was measured separately. Based on PCNA-GFP fluorescence patterns, duration of each cell cycle phase was analyzed for epidermal cells in hair cell lineage.

**GUS staining**. Tissue was pre-fixed in 90% (v/v) acetone for 20 min on ice, rinsed with 100 mM sodium phosphate solution (pH 7.0), and transferred to staining solution (100 mM sodium phosphate buffer pH 7.0, 0.1% Triton-X-100, 0.5 mM potassium ferrocyanide, 0.5 mM potassium ferricyanide, 10 mM EDTA, 0.5 mg mL$^{-1}$ 5-bromo-4 chloro-3-indolyl-β-D-glucuronic acid) and incubated at 37 °C for 12 h. GUS-stained samples were then transferred to 70% ethanol, incubated to remove chlorophyll and then photographed or used for microscopic observation.

**Reporting summary**. Further information on research design is available in the Nature Research Reporting Summary linked to this article.

## Data availability

RNA-Seq data for WT, *scl28*, and *atsmos1* can be accessed from the DDBJ database under accession number DRA012786. ChIP-Seq data of SCL28 and AtSMOS1 can be accessed at Gene Expression Omnibus database under accession number GSE183209. Arabidopsis mutants and transgenic lines, as well as plasmids generated in this study are available from the corresponding author upon reasonable request. Source data are provided with this paper.

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

## Acknowledgements

The authors thank Kyoko Kato, Akiko Nakanishi, Natsuko Ono, Akiko Yamamoto, Chikako Inoue, Nanako Ishibashi, Satoko Nasu, Yuiko Tachikawa, Asako Segawa, Ayami Furuta, Tomomi Shinagara, Ayako Nakamura, and Ayumi Yamada for technical assistance. This study was funded by The Japan Society for the Promotion of Science KAKENHI (20H05408 and 17H03696 to M.I., 20H05911 and 20H05905 to K.S.), Japan Science and Technology Agency (JST grant number JPMJPF2102 to M.I.) and the Institut Universitaire de France (to M.B.). Y.H. was supported by China Scholar Council fellowships (201806690005).

## Author contributions

M.I. conceived the study; C.R., M.B., Z.M., K.S., T.Y., and M.I. designed the experiments; M.Imamura., K.M., T.I., C.B., M.G., D.L., Y.H., T.S., K.Y., H.T., and Y.N. carried out the experiments; J.A., Takamasa. S. analyzed the data; M.I., L.B., Z.M., and M.B. wrote the paper. H.T. and Y.N. contributed equally.

## Competing interests

The authors declare no competing interests.
