## [Peer Review File · Nature Communications]

A hierarchical transcriptional network activates specific CDK inhibitors that regulate G2 to control cell size and number in ArabidopsisREVIEWER COMMENTS

Reviewer #1 (Remarks to the Author):

The work presented in this manuscript is a significant contribution to our understanding of the plant cell cycle, as well as providing a basis for comparison with animal and fungal cell cycles. Transcriptional control of the G2/M phase of the cell cycle is only partially understood. In both plants and animals, a protein complex termed the DREAM complex plays an important role in both G1/S and G2/M transcription via both E2F and MYB3R transcription factors that are part of this multi-protein complex. Because this complex is found in both animals and plants, organisms that independently evolved multicellularity, this complex must have deep roots in the evolution of the eukaryotic cell cycle. The dynamic behavior of the DREAM complex is complex, however, and the plant complex differs in important ways from the complex found in animals. In this manuscript, the authors have identified the transcription factor SCL28 as a downstream target of the MYB3R proteins, and have shown that SCL28 negatively regulates the G2/M transition and regulates cell cycle length, cell number and cell size during organ growth via the transcription of several SMR-type CDK inhibitors. Importantly, though buried in Supplementary figures, they show that only MYB3Rs, and not other DREAM components such as E2Fs or RBR, regulate SCL28 transcription. They also identify AtSMOS1 as a necessary dimerization partner for SCL28 that is regulated independently of the MYB3Rs. This work represents a significant advance in our understanding of the transcription network regulating G2/M in relation to organ growth, and gives new insight into the function of the plant DREAM complex. Multiple lines of evidence support each of these points, making the claims in the manuscript very convincing. Additionally, the writing is clear and concise.

I do have a few suggestions that could improve the manuscript. First, the authors focus on the effects of SCL28 and SMRs on palisade mesophyll cell size and number. However, the work of Tsukaya and colleagues (Tsukaya and Matsunaga, *Plant Morphology* 29:87, 2017; Katagiri et al., *Development* 2016, doi:10.1242/dev.130021) has shown the relationship between cell size and ploidy is quite different between the epidermis and mesophyll. I would be very curious to know what happens to cell number and cell area for epidermal cells, in addition to mesophyll, in *scl28* mutants.

Second, there are a few minor points that could improve the manuscript:

p. 4, second to last line, the meaning of “almost diminished” is inexact and unclear, and alternative phrasing should be substituted.

p. 9, toward the end of the first paragraph, “proSMR2::SMR-GFP” should be proSMR2::SMR2-GFP (the second “2” is missing).

p. 15, last sentence in the second paragraph, starting “It appears that SCL28 acts independently of DREAM mechanisms,” seems to me not entirely clear. The authors may wish to rephrase or even expand the point, which is an important one.

Reviewer #2 (Remarks to the Author):

Both cell cycle regulation and cell size control are fields of interest in developmental biology across organisms. Our understanding cell size control in plants has increased vastly in the last few years, however, the G2/M transition in plants, and particularly the mechanism by which cell size information is integrated into this mechanism, remain poorly understood. In this paper Nomoto et al. identify a regulatory module which links a target of MYB3R expression (SCL28) to meristematic cell size regulation via the transcription control of several SMR genes including SMR2, which negatively regulate the activity of CDKs. Although SCL28 has previously been shown to be expressed during G2-M and linked to cell size control (Goldy et al 2021, PNAS 118 (6) e2005256118; <https://doi.org/10.1073/pnas.2005256118>) Nomoto et al provide several important additional mechanistic insights:

- SCL28 binds AtSMOS1 to form a heterodimer that activates gene transcription
- SMR1, SMR2, SMR6, SMR8, SMR9, SMR13, and SMR14 are targets of SCL28/AtSMOS1
- SMR genes mediate the effect of SCL28/AtSMOS1 on cell size.

The authors use a combination of bioinformatics, live cell imaging and classic genetics to produce multiple lines of evidence for each claim and the paper is well written and interesting.

I have a few specific comments about the methods/data presented:

Regulation of SCL28 by DREAM complex. The authors conclude that SCL28 is not regulated by the DREAM complex on the basis of no change in SCL28 transcript in qPCR experiments conducted in mutants of DREAM components. Some of the qPCR results have very large error bars (particularly e2fb-1 and e2fc-1). I think it is difficult to conclude anything for these particular genes. Also, since the authors identified E2F binding sites in the AtSMOS1 promoter it would be interesting to see quantification of AtSMOS1 in these backgrounds.

Dimerisation – The authors include several experimental lines of evidence for dimerization including BiFC and yeast-II hybrid. These are appropriate and well established techniques, but can be prone to non-specific association. This has been controlled for in the Yeast-II Hybrid, but in the case of the BiFC

experiment the authors do not present a negative control in which both halves of the YFP protein were present, but no interaction was detected. This control would strengthen the interpretation.

Identification of shared targets – The authors use a transcriptional analysis of *scl28*, *smos1* and *SCL28OE* to identify targets of the *SCL28/AtSMOS1* heterodimer. In general, I find that the number of genes that overlap seems quite low – not only in terms of those genes that are shared targets of *SCL28* and *AtSMOS*, but also the number of genes that are for example down regulated in *scl28* and upregulated in *SCL28OE*. More detail is needed regarding the methods used to identify these genes is needed in order to understand why these numbers are low and whether this experiment supports the theory that the two proteins are co-ordinately regulating a suite of genes.

KRPs – the authors state that none of the KRP genes examined showed change in expression, but the level of *KRP2* and *KRP4* are shown to be significantly different in Supplementary Fig 10. Given that only *SMR2* was picked up in the initial transcriptomic analysis (with changes in others subsequently detected by qPCR), were the KRP results also confirmed by qPCR?

Sample sizes – in general the sample sizes could be reported more clearly for example, in qPCR data does $n=3$ refer to the number of technical or experimental repeats, and in Fig 2 d-g does n refer to cells or leaves in each case?

Minor Points:

Results: First paragraph, first line “By analyzing transcripts specific in mitotic cells, we have sience identified an Arabidopsis GRAS family transcription factor” – there is a typo here – I am not sure what this should have been.

Page8 – last sentence “as well as 26 genes affected in an opposing way within each line (Fig 5b)” this description could be clearer. It would be easier to understand if the it was listed (upregulated in *scl28* and *smos1* and downregulated in *SCL28OE*)

Point-by-point response to reviewers' comments

Response to reviewer#1

Comment 1

First, the authors focus on the effects of SCL28 and SMRs on palisade mesophyll cell size and number. However, the work of Tsukaya and colleagues (Tsukaya and Matsunaga, *Plant Morphology* 29:87, 2017; Katagiri et al., *Development* 2016, doi:10.1242/dev.130021) has shown the relationship between cell size and ploidy is quite different between the epidermis and mesophyll. I would be very curious to know what happens to cell number and cell area for epidermal cells, in addition to mesophyll, in *scl28* mutants.

Our response:

We have conducted additional experiments for kinematic analysis of leaf growth focusing on epidermal cells, and added new Supplementary Fig. 4 to the revised manuscript. We used leaves at 6, 9 and 12 d after germination, and analyzed size and number of epidermal cells in first leaf pairs. We found that the cell size regulatory role of SCL28 in mesophyll cells also applies to epidermal cells; the average size of pavement cells is smaller in *scl28* and larger in *SCL28^{OE}* compared to wild type throughout the period of the experiment. Similarly to mesophyll cells, we also found that the number of epidermal cells per leaf increased in *scl28* and decreased in *SCL28^{OE}* compared to wild type. Thus we conclude that the role of SCL28 is universal across a wide variety of cell types. We added the following sentence to the revised manuscript (line 4-7, page 6).

Sentence added: We also analyzed cell size and number of epidermal pavement cells during leaf development in *scl28* and *SCL28^{OE}* lines and found the response of this cell type to altered SCL28 activity was comparable to what we have shown for palisade cells (Supplementary Fig. 4).

Comment 2

Second, there are a few minor points that could improve the manuscript:

p. 4, second to last line, the meaning of “almost diminished” is inexact and unclear, and alternative phrasing should be substituted.

Our response:

We have rephrased the sentence as below.

Rephrased sentence: GUS reporter activity driven by *SCL28* promoter decreased significantly in the *myb3r1/4* double mutant and became essentially undetectable when the MSA elements in the *SCL28* promoter had been deleted (Fig. 1d).

Comment 3

p. 9, toward the end of the first paragraph, “proSMR2::SMR-GFP” should be proSMR2::SMR2-GFP (the second “2” is missing).

Our response:

We have corrected.

Comment 4

p. 15, last sentence in the second paragraph, starting “It appears that SCL28 acts independently of DREAM mechanisms,” seems to me not entirely clear. The authors may wish to rephrase or even expand the point, which is an important one.

Our response:

We agree with the reviewer that this sentence was not clear. What we meant that while SCL28 solely acts in proliferating cells, the RBR/E2F/DREAM pathway also regulates cell cycle exit and might regulate cell size through this mechanism.

Rephrased sentence: Whether the RBR/E2F/DREAM pathway directly regulates the cell size in proliferating cells or its cell size effect is through the regulation of cell cycle exit, remains to be established. It will be also important to know how the RBR/E2F/DREAM and the SCL28/AtSMOS1 pathways are coordinated.

Response to reviewer#2

Comment 1

Regulation of SCL28 by DREAM complex. The authors conclude that SCL28 is not regulated by the DREAM complex on the basis of no change in SCL28 transcript in qPCR experiments conducted in mutants of DREAM components. Some of the qPCR results have very large error bars (particularly e2fb-1 and e2fc-1). I think it is difficult to conclude anything for these particular genes. Also, since the authors identified E2F binding sites in the AtSMOS1 promoter it would be interesting to see quantification of AtSMOS1 in these backgrounds.

Our response:

We have repeated this experiment by quantifying *SCL28* mRNA levels by qRT-PCR using newly harvested seedlings. The results are added in revised Supplementary Fig. 1b by replacing the old

version. We confirmed that there is no significant difference in the expression of *SCL28* in any of the *e2f*, *rbr* and *tcx* mutant lines compared to WT.

As requested, we also quantified mRNA levels of *AtSMOS1* using the same RNA samples, and found it to be significantly (around 1.5 fold) upregulated in *e2fa e2fb e2fc* triple mutant, suggesting that *AtSMOS1*, that possess an E2F motif in its promoter regions, is actually regulated by E2Fs. We also showed by ChIP-qPCR experiment that E2FB as well as RBR bind to the *AtSMOS1* promoter *in vivo*. Our new data was included in Supplementary Fig. 9 in the revised manuscript. Despite the significant effect of E2Fs on *AtSMOS1* transcription, our qRT-PCR analysis did not detect significant upregulation of *AtSMOS1* in *rbr* and *tcx5 tcx6*. This result may suggest that regulation of *AtSMOS1* by E2Fs does not exclusively depend on the DREAM complex, and that effect of RBR and DREAM on E2F may be developmental specific. We have added our interpretation in lines 12-17, page 8 in the revised manuscript.

Comment 2

Dimerisation – The authors include several experimental lines of evidence for dimerization including BiFC and yeast-II hybrid. These are appropriate and well established techniques, but can be prone to non-specific association. This has been controlled for in the Yeast-II Hybrid, but in the case of the BiFC experiment the authors do not present a negative control in which both halves of the YFP protein were present, but no interaction was detected. This control would strengthen the interpretation.

Our response:

We have conducted a new BiFC experiment with a complete set of negative controls in which two expression plasmids were transfected in the following three combinations.

1) *AtSMOS1*-nYFP and cYFP only, 2) nYFP only and *SCL28*-cYFP, 3) nYFP only and cYFP only

In this experiment, we could confirm clear YFP fluorescence only when both *AtSMOS1*-cYFP and *SCL28*-nYFP are transfected into protoplasts. In the revised manuscript, we replaced old Fig. 4b with this new data set.

Comment 3

Identification of shared targets – The authors use a transcriptional analysis of *scl28*, *smos1* and *SCL28OE* to identify targets of the *SCL28/AtSMOS1* heterodimer. In general, I find that the number of genes that overlap seems quite low – not only in terms of those genes that are shared targets of *SCL28* and *AtSMOS*, but also the number of genes that are for example down regulated in *scl28* and upregulated in *SCL28OE*. More detail is needed regarding the methods used to identify these genes is needed in order to understand why these numbers are low and whether this experiment supports the theory that the two proteins are co-ordinately regulating a suite of genes.

Our response:

We defined differentially expressed genes (DEGs) as those with adjusted *P*-value < 0.05 in our statistical analysis of the expression data from three biological replicates. The definition of DEG was added in the main text (lines 4-5 from the bottom, page 8), and also in the legend for Fig. 5a in the revised manuscript.

Reviewer#2 also pointed out the relatively small overlap between DEGs in *scL28* and *SCL28^{OE}*, and in *scL28* and *atsmos1*.

1) Interpretation for small overlap between DEGs in *scL28* and *SCL28^{OE}*

In *SCL28^{OE}* plants, we found a severe dwarf phenotype due to strong negative effect by *SCL28^{OE}* on cell division, while the *scL28* mutant line is macroscopically normal. Such severe phenotype is expected to be associated with significant changes of overall transcriptome, in which the majority of DEGs may be due to indirect effects of *SCL28^{OE}*, but not through its direct targets. In fact, 25% of downregulated genes in *scL28* are shared in those upregulated in *SCL28^{OE}*, whereas only 8% of upregulated genes in *SCL28^{OE}* is downregulated in *scL28* (please see Fig. 5a).

2) Interpretation for small overlap between DEGs in *scL28* and *atsmos1*

In our ChIP-seq experiment, we identified a much larger number of target genes for AtSMOS1 (4,287 genes) than SCL28 (463 genes) (please see Fig. 6a). When focusing on the target genes of SCL28, 66% of them are shared targets of AtSMOS1 targets. In contrast, if we focus on the AtSMOS1 targets, only 7% are shared with SCL28 targets. This result led us to consider that AtSMOS1 should have a broader range of function which are independent of SCL28. This is also consistent with its broad expression pattern as was shown in proAtSMOS1::AtSMOS1-GFP plants. Conversely, we reasoned, based on our genetic analysis, that the majority, if not all of SCL28 function are mediated by its interaction with AtSMOS1. Therefore, relatively small overlap between downregulated genes in *scL28* and *atsmos1* may be due to presence of large number of AtSMOS1 targets that may be regulated independently of SCL28. This explanation for the small overlap was added to the main text of the revised manuscript (lines 4-6, page 10).

Comment 4

KRPs – the authors state that none of the KRP genes examined showed change in expression, but the level of KRP2 and KRP4 are shown to be significantly different in Supplementary Fig 10. Given that only SMR2 was picked up in the initial transcriptomic analysis (with changes in others subsequently detected by qPCR), were the KRP results also confirmed by qPCR.

Our response:

We conducted additional qRT-PCR experiment to examine the expression of all *KRP* genes in *scL28*,

atsmos1, and *SCL28*^{OE}. In contrast to SMRs, the expression change of some of the *KRPs* were not consistent to what were expected for the *scl28* and *atsmos1* mutants in relation to their action as dimer, that is simultaneous down regulation, while induction by *SCL28*^{OE}. This suggests that *KRPs* are not direct targets of the *SCL28/AtSMOS1* heterodimer, but might be indirectly affected in these lines. This was further supported by our ChIP-seq data showing that none of the *KRP* genes were identified as common targets of *SCL28* and *AtSMOS1*. These explanations were added in the main text (lines 15-21 page 9, and lines 23-25 page 10), and the expression data from new qRT-PCR experiment was added as Supplemental Fig. 12b in parallel with the previous data from transcriptome analysis (Supplemental Fig. 12a).

Comment 5

Sample sizes – in general the sample sizes could be reported more clearly for example, in qPCR data does n=3 refer to the number of technical or experimental repeats, and in Fig 2 d-g does n refer to cells or leaves in each case?

Our response:

We have added detailed explanations for sample size and type of replicates in all corresponding Figure legends, which include Fig. 1a, 1c, 2d-2g, 2h, 2i, 3a-3d, 4d, 4f, 4h, 4j, 5c, 5d, 6f, 7a, 7e, 7g, 8b, 8c, and Supple Fig. 1b, 4a, 4c, 9a, 9b, 12a, 12b, 18b, 18d in revised manuscript.

Comment 6

Results: First paragraph, first line “By analyzing transcripts specific in mitotic cells, we have sience identified an Arabidopsis GRAS family transcription factor” – there is a typo here – I am not sure what this should have been.

Our response:

We have corrected this typo by deleting “sience”.

Comment 7

Page8 – last sentence “as well as 26 genes affected in an opposing way within each line (Fig 5b)” this description could be clearer. It would be easier to understand if the it was listed (upregulated in *scl28* and *smos1* and downregulated in *SCL28*^{OE})

Our response:

We have rephrased the sentence as follows by fully explaining gene category (lines 2-3 from the

bottom, page 8).

Rephrased sentence: -and identified 21 genes that are significantly (adj *P*-value less than 0.05) downregulated in both *sc128* and *atsoms1*, and upregulated in *SCL28*^{OE} (Fig. 5a and Supplementary Fig. 8a), as well as 26 genes upregulated in both *sc128* and *atsoms1*, and downregulated in *SCL28*^{OE} (Fig. 5b and Supplementary Fig. 8b).

REVIEWERS' COMMENTS

Reviewer #1 (Remarks to the Author):

The authors have provided clear and convincing evidence that SCL28 also affect epidermal cell size and number, which was my only serious scientific question. The authors have addressed all of my minor comments as well. Finally, in my opinion, the authors have been successful in addressing the comments of the other reviewer as well.

Reviewer #2 (Remarks to the Author):

I am satisfied that the concerns I raised in my original review have been addressed through the inclusion of new data and additional information/discussion in the text.

As previously stated, this is an important and well written paper that will be of significance to the field. I have no further queries for the authors.